**Subject Category:**
Biology (whole organism)

ecology/environmental science

zooplankton, stoichiometry, body-size, intraspecific variation, phosphorus

**Author for correspondence:**
Rachel Hartnett
e-mail: rhartnett@ou.edu

# Variation in life-history traits among *Daphnia* and its relationship to species-level responses to phosphorus limitation

## Rachel Hartnett

Department of Biology, Program in Ecology and Evolutionary Biology, The University of Oklahoma, 730 Van Vleet Oval, Room 314, Norman, OK 73019, USA

RH, 0000-0003-1743-4430

Currently organisms are experiencing changes in their environment at an unprecedented rate. Therefore, the study of the contributions to and responses in traits linked to fitness is crucial, as they have direct consequences on a population's success in persisting under such a change. *Daphnia* is used as a model organism as the genus contains keystone primary consumers in aquatic food webs. A life-history table experiment (LHTE) using four species of *Daphnia* was conducted to compare variation in life-history traits among species across two different environmental conditions (high and low phosphorus availability). Results indicate that the food quality environment had the most impact on life-history traits, while genetic contributions to traits were higher at the species-level than clonal-level. Higher trait variation and species-level responses to P-limitation were more evident in reproductive traits, while growth traits were found to be less affected by food quality and had less variation. Exploring trait variation and potential plasticity in organisms is increasingly important to consider as a potential mechanism for population persistence given the fluctuations in environmental stressors we are currently experiencing.

## 1. Background

With increasing environmental stress, many suites of organismal traits are expected to experience strong selection, with life-history traits potentially being among the most impacted [1,2]. Life-history traits have a direct link to fitness, as an organism's success is built upon an ability to grow to reproductive age, the timing of reproduction events, as well as cumulative reproductive output.

Therefore, life-history theory has established direct associations between a population's environment and life-history trait evolution [3,4].

Food stress has been shown to create a variety of life-history trait effects in organisms, which include longer developmental time, decreases in body size and lowered fecundity [5,6]. Food stress can be experimentally manipulated by decreasing a limiting resource. In many freshwater lentic systems, phosphorus (P) is one of the limiting nutrients [7,8], with anthropogenic inputs of P in aquatic systems forcing rapid change in zooplankton populations [9]. P-limitation (i.e. low food quality) has effects on *Daphnia* life-history traits such as growth, reproduction and senescence (e.g. [10,11]). Members of the genus *Daphnia* (Cladocera: Anomopoda) have one of the highest P contents among zooplankton, so they are predicted to be more responsive to P-limitation compared to other zooplankton taxa [12]; however, P content alone is not sufficient to predict species-level responses in growth [13].

*Daphnia* have a cyclically parthenogenic life cycle, which includes bouts of asexual reproduction under good growing conditions, and sexual reproduction during times of food stress, changes in photoperiod, and crowding cues [14]. This creates the unique advantage of establishing multiple clonal lineages in a population leading to the maintenance of high genetic variation in many natural *Daphnia* populations [15–17]. In addition, researchers have found strong clonal responses to predator cues [18,19], nutrient limitation [20,21], habitat selection [22] and toxins [23,24]. Intraspecific genetic variation has been shown to have effects on population-level processes like colonization [25,26], coexistence [27] and predation [28]. For *Daphnia*, clonal diversity is better maintained under P-limitation [29]; therefore, clonal variation may significantly contribute to overall genetic variation among *Daphnia* species.

Under the hypothesis of environmental buffering, influential life-history traits should have minimal trait variation, as fitness would be heavily dependent on minimal change within important vital rate constraints [30]. Over multiple life-history tables, Pfister [30] showed that traits with the highest variation had the least response to temporal change. This demonstrates a potential trade-off between trait variation and species-level responses to environmental change, where traits that are more vital to fitness would show less trait variation. Life-history traits vary in the amount of variation seen under food stress; P-limitation has been shown to differentially affect somatic growth rates and reproductive investment like in egg size and abortion rates within *Daphnia* due to differing nutritional requirements [31]. Alternatively, trait variation and species-level responses to environmental changes could be positively correlated, as traits with low variation may have a lower range of trait values that are measurable in species-level responses (i.e. a floor effect), and it has been documented that more intraspecific trait variation also leads to higher variation in intraspecific responses across environments (e.g. [32]). This trend between intraspecific trait variation and intraspecific responses could translate to species-level responses as well.

Another mechanism to mitigate environmental effects is an organism's body size. *Daphnia*'s physiology allows them to alter filtering rates under different food quality environments [33], although there are phylogenetic constraints. Jeyasingh [34] suggested that evolution should favour more responsive physiologies for smaller organisms in order to counter frequent shifts in nutrient limitation. Smaller aquatic organisms use phosphorus at higher rates [35], which means that they would experience P-limitation more frequently. This could lead to smaller-bodied species becoming more plastic than larger-bodied species.

This present study aims to address the following: (i) how much does the environment versus genetic (taxonomic) identity contribute to life-history trait variation? Do clonal lines contribute more to trait variation than species identity? (ii) Do smaller species exhibit more plastic physiologies? And (iii) what is the potential relationship between intraspecific variation and species-level responses to food stress?

# 2. Methods

## 2.1. Study organism

*Daphnia* is a cosmopolitan genus [36,37]. Three clonal lineages from each of four different *Daphnia* species (*D. magna*, *D. mendotae*, *D. obtusa* and *D. pulex*) were collected from a variety of laboratory stocks (table 1). These clonal lineages span the three subgenera of *Daphnia*, ranging across North America and Europe, and come from various aquatic habitats (table 1). *Daphnia magna* clones used in this study originated from South Dakota, Finland and Germany from a spectrum of habitats. The South Dakotan clone (MA3) came from a permanent lake, a shallow (less than 2 m) prairie pothole [21]. MA2 and MA1 are both inbred lines from an original genetic cross between a Finnish clone and a German clone. MA2

**Table 1.** Species list of *Daphnia* populations used in the life-history table experiment (LHTE).

| species | clone | location | habitat type |
| --- | --- | --- | --- |
| *D. magna* | MA1 | Munich, Germany | semi-permanent lake [38] |
| *D. magna* | MA2 | Tvärminne, Finland | ephemeral rockpool [38] |
| *D. magna* | MA3 | South Dakota, USA | shallow, permanent lake [21] |
| *D. obtusa* | OB1 | Oklahoma, USA | pond |
| *D. obtusa* | OB2 | Illinois, USA | pond |
| *D. obtusa* | OB3 | Missouri, USA | pond |
| *D. pulex* | PX1 | Illinois, USA | shallow pond [20] |
| *D. pulex* | PX2 | Illinois, USA | shallow pond [20] |
| *D. pulex* | PX3 | Illinois, USA | shallow pond [20] |
| *D. mendotae* | ME1 | Minnesota, USA | permanent lake |
| *D. mendotae* | ME2 | Minnesota, USA | permanent lake |

was inbred for three generations and MA1 was inbred for one generation (Dieter Ebert, Switzerland, 2013, personal communication). The environment of the parental clones include a Finnish clone from an ephemeral pond with desiccation in spring/summer and freezing during autumn/winter and a clone from a German semi-permanent pond, with freezing in the winter [38]. In addition, *D. pulex* and *D. obtusa* clones came from temporary ponds in the US Midwest, while *D. mendotae* came from permanent lakes in the US Midwest. One *D. mendotae* clone (ME3) experienced high levels of mortality early on in the experiment, and was subsequently dropped from the analyses. These contrasting environments have created very different evolutionary trajectories for these species. However, a couple of caveats should be noted: these lineages have been adapted to laboratory conditions and may have different capacities in variation and response than animals collected directly from the field, and a potential confounding issue due to inbreeding could affect the variation and response for two of the three *D. magna* clones (MA1 and MA2).

## 2.2. Experimental design

Clonal lineages were maintained as separate populations in 900 ml jars, with regular and plentiful feeding using the chemostatically cultured green algae, *Scendesmus acutus*, at a constant 20°C in COMBO media [39]. These stock cultures maintained an approximate density of 12–30 adults $l^{-1}$. A small amount of cetyl alcohol (approx. 10 mg) was added to act as a surfactant to prevent animals from being trapped in the air–water interface. Stock cultures and experimental animals received equal amounts of 24 h incidental ambient lighting. Maternal lines for experimental animals were raised individually in 60 ml jars with 50 ml of COMBO and fed daily with 1 mg C $l^{-1}$ of *S. acutus* that was grown in nutrient-rich conditions (i.e. C : P, ~100 : 1). The volume of *S. acutus* added each day for 1 mg C $l^{-1}$ was calculated by measuring the absorbance of chlorophyll at wavelengths 660 nm and 740 nm using a spectrophotometer (Spectronic®20 Genesis, Madison, WI) and using a calibrated chlorophyll-carbon curve [40]. Females were monitored every 24 h, and first and second clutches were removed. Experimental animals ($N = 20$ per clone) were taken from the third or later clutches of these individually raised maternal lines within 24 h to reduce maternal effects [41]. Experimental animals needed to be pooled from multiple mothers in order to reach the appropriate sample size.

Starting on 17 August 2013, an initial body-length measurement (i.e. start length) of each experimental animal was taken using a MOTICAM 2300 digital camera and software system (Motic®, S-05165) mounted to an Olympus BX51 compound dissecting microscope. Length measurements were taken from the top of the eyespot to the base of the core body, right above the top of the tail-spine. The tail-spine is known to be morphologically plastic depending on environmental conditions and it was not measured with core length due to potential confounding length measurements. Then, experimental animals were placed individually into 60 ml glass jars with 50 ml of COMBO at 20°C and were divided into two environmental conditions, high and low phosphorus ($N = 10$ per clonal line for each environmental treatment). Nutrient media was prepared using a modification of COMBO

[39], with additions of sodium nitrate and potassium phosphate. Animals under a high phosphorus (HiP) feeding regime were fed daily with 1 mg C $l^{-1}$ of *S. acutus* that was grown in nutrient-rich conditions (i.e. 85.01 mg $l^{-1}$ sodium nitrate and 8.71 mg $l^{-1}$ potassium phosphate and C : P ~100 : 1). A low phosphorus (LoP) feeding regime consisted of daily 1 mg C $l^{-1}$ feeding of *S. acutus* grown in nutrient-poor conditions (i.e. 42.5 mg $l^{-1}$ sodium nitrate and 0.87 mg $l^{-1}$ potassium phosphate and C : P ~750 : 1). These food conditions should not be limiting in quantity [42]. Experimental animals were transferred every 2 days to fresh jars in order to avoid carbon (detrital) accumulation that could differentially affect resource availability based on inter-/intra-specific variation of filtering rates. Experimental animals were monitored daily and size was measured again at maturation, when first egg development was seen (i.e. age at maturation and length at maturation). Clutch size was recorded daily, as well as images for neonate body-lengths ($N \leq 5$ neonates per clutch in order to reduce small-clutch bias). Number of clutches, clutch size and mean neonate length (termed mean clutch length) were calculated from these daily recorded measurements. Dead experimental animals were measured on the day of death. The experiment ran for 28 days, and at the end of this period, experimental animals were measured (i.e. end length), as described above.

## 2.3. Statistical analyses

All analyses were conducted using R [43] unless specified otherwise in the methods. Required packages for this analysis included: dplyr [44], ggplot2 [45], ggpubr [46] and Hmisc [47]. Individuals (replicates) were dropped from the analysis if they died within 5 days of the start of the experiment to prevent bias from missing data in reproductive measurements. Data were screened for outliers, and cases were removed if they were ±2 standard deviations from a mean value. Mortality accounts for less than 5% of the study, and a $\chi^2$ contingency analysis was conducted between species and treatment to ensure that survivorship was independent of species identity or treatment. A MANOVA was conducted to measure the effects of genetic (species, clonal) and environmental (phosphorus treatment) contributions on life-history traits (length at maturation, age at maturation, end length, mean clutch length, mean clutch size and number of clutches). The start length of individuals was used as a covariate, as well as maternal line; maternal effects are common among daphniid studies [48], so maternal line was also used as a potential confounding variable. Multivariate normality was checked for the dataset using *post hoc* residuals from the MANOVA, and the proportion of variance explained for each main effect was calculated from Wilks' lambda, where partial $\eta^2 = 1 - \Lambda^{1/s}$; $\Lambda$ is Wilks' lambda for the main effect and $s = \min(6, \text{d.f.}_{\text{effect}})$ [49]. The MANOVA was conducted using SPSS (Version 20, IBM).

To identify differences in species-level responses to food stress, multiple linear regressions were conducted for each trait, with food treatment and species as factors. The significance-level was adjusted using the Bonferroni correction (alpha = 0.008). Intraspecific variation was measured for each species by coefficients of variation (COVs). COVs were calculated from CV = $s/\gamma$, where $s$ = standard deviation and $\gamma$ = mean of the particular life-history trait for that species. Thus, COVs measure the amount of variation within the species for each trait. Species-level responses to food stress were quantified by using the differences in log-transformed values between phosphorus treatments [50]: species-level response per trait = $\ln(\text{trait}_{\text{HIP}}) - \ln(\text{trait}_{\text{LOP}})$ for each species. Individual *t*-tests were conducted on species-level responses and COVs, respectively, between the larger species (*D. magna* and *D. pulex*) compared to the smaller species (*D. mendotae* and *D. obtusa*). A Pearson correlation between COVs and these species-level response values were also carried out to test the relationship between trait variation and the magnitude of response to food stress. Separate correlations were conducted per species and per trait; the significance-level was adjusted using the Bonferroni correction (alpha = 0.002).

## 3. Results

Mortality in this study was independent of species identity and food treatment ($\chi^2$ contingency, d.f. = 3, $\chi^2$ = 0.1361, $p$ = 0.9872). Descriptively, under low-phosphorus (LoP) conditions, all clonal lines of all species showed smaller sizes both at first reproduction, and at the end of the experiment. Similarly, under LoP, clones exhibited delayed onset of reproduction and had smaller clutch sizes. The number of clutches varied per clone, as well as their mean clutch length (see electronic supplementary material, S1). Visual inspection of trait variation mapping using a PCA indicated that species showed more change in reproductive traits rather than size traits between food treatments (see electronic supplementary material, S2).

**Table 2.** Factorial MANOVA scores. Main effects and two-way interactions from a factorial MANOVA are shown here. 'Food quality' indicates the main effect of the food treatment manipulation (high phosphorus—HiP/low phosphorus—LoP). 'Species' indicates the main effect of species on the response variable. 'Clone' indicates the level of effect at the clonal-level, nested within species, on the response variable. Body length (mm) at the start of the experiment, the mother of the experimental animals, and time blocks were used as covariates. Two-way interactions were also tested.

| source of variance | Wilks' Lambda | d.f.1 | d.f.2 | multivariate $F$ | partial $\eta^2$ |
|---|---|---|---|---|---|
| start length (covariate) | 0.909 | 6 | 146 | 2.42 | |
| maternal effects (covariate) | 0.986 | 6 | 146 | 0.343 | |
| time (covariate) | 0.925 | 6 | 146 | 1.983 | |
| food quality | 0.150 | 6 | 146 | 137.409*** | 0.85 |
| species | 0.064 | 18 | 413 | 37.863*** | 0.60 |
| clone | 0.210 | 42 | 688 | 6.463*** | 0.23 |
| species × food | 0.167 | 18 | 413 | 20.250*** | |
| clone × food | 0.195 | 42 | 688 | 6.843*** | |

\*\*\*$p < 0.0001$.

The MANOVA showed that both genetic factors (species and clone) as well as environmental factors (food treatment) significantly affected life-history traits (table 2). The length of individuals at the start, the blocking effect of time and maternal effects did not significantly affect life-history traits (table 2). The proportion of variance explained can be used as a proxy for the magnitude of effect size metric [49]. Using the partial $\eta^2$ statistic as a relative effect size metric, food quality environment explained the greatest proportion of variance (d.f.$_{effect}$ = 1), while genetic factors explained less; species identify (partial $\eta^2 = 0.60$, d.f.$_{effect}$ = 3) explained more than the clonal level (partial $\eta^2 = 0.23$, d.f.$_{effect}$ = 7) (table 2). Therefore, the environment had a stronger main effect than either genetic component, with species identity having a stronger effect than clonal identity.

The size of the species did not show significant differences in species-level responses (two-samples $t$-test, $t = 0.75$, d.f. = 22, $p > 0.1$). However, they did show significant differences in the amount of intraspecific variation among traits, with larger species showing more variation than smaller species (two-samples $t$-test, $t = 8.41$, d.f. = 22, $p < 0.01$).

All life-history traits were impacted by both species identity and the food quality level (length at maturation, adjusted $R^2 = 0.921$; age at maturation, adjusted $R^2 = 0.670$; end length, adjusted $R^2 = 0.752$; mean clutch length, adjusted $R^2 = 0.146$; mean clutch size, adjusted $R^2 = 0.708$; number of clutches, adjusted $R^2 = 0.464$; all adjusted $p < 0.01$). Effects of food quality were detected across all species for some reproductive traits (length at maturation and clutch size), while mean clutch length only had effects for larger species *D. magna* and *D. pulex* (figure 1). *Daphnia mendotae* was the only species that had an effect with food quality treatment on the length of individuals at the end of the experiment (figure 1).

Intraspecific trait variation and species-level responses were significantly positively correlated ($r^2 = 0.89$, $t = 9.60$, d.f. = 22, $p < 0.01$; figure 2). Correlations conducted per species found that *D. magna* and *D. obtusa* had significant correlations between intraspecific trait variation and species-level responses (*D. magna*: $r^2 = 0.94$, $t = 5.51$, d.f. = 4, $p < 0.001$; *D. obtusa*: $r^2 = 0.99$, $t = 12.151$, d.f. = 4, $p < 0.001$). There was no significant correlation detected when analysing correlations per trait.

# 4. Discussion

## 4.1. The environment contributed most to life-history traits, and species identity explained greater trait variation than genotype

A keystone of evolutionary theory is that trait variation can be separated into genetic and environmental components; when a selection pressure is imposed on a trait, the environmental variance will dominate over genetic variance due to the reduction in genetic variance in the selection process [51]. The food quality available in the environment had the largest effect on life-history traits, accounting for 85% of

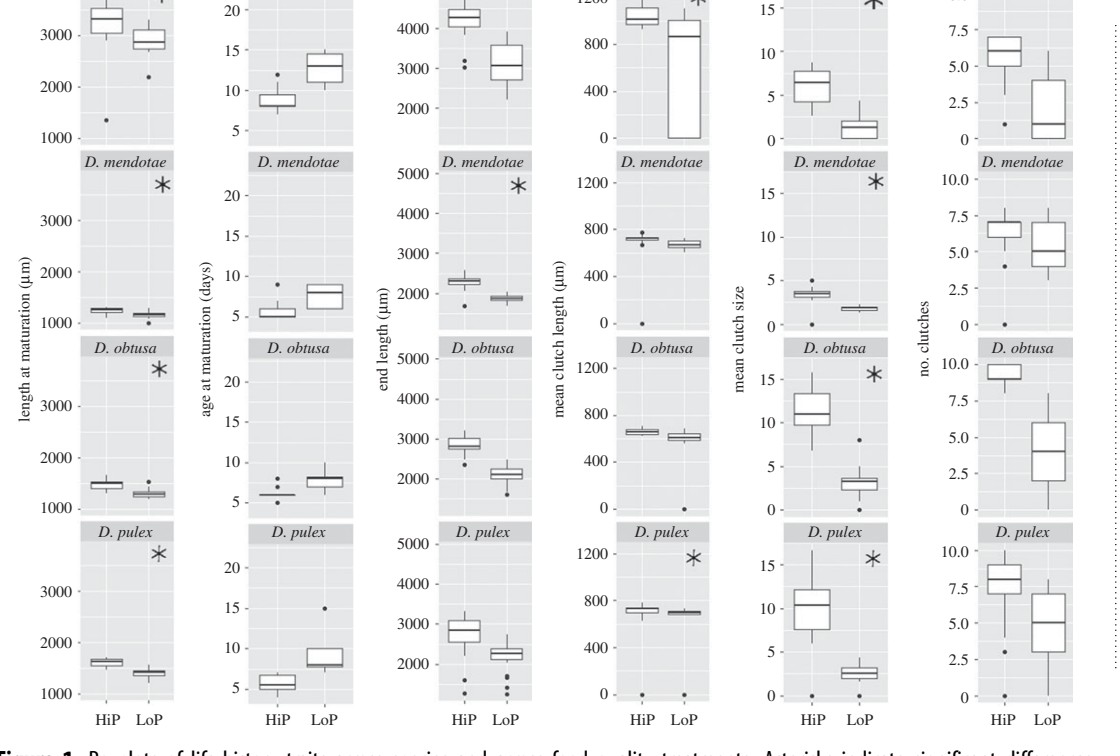

**Figure 1.** Boxplots of life-history traits across species and across food quality treatments. Asterisks indicate significant differences between treatments were detected using linear regressions, with adjusted significance levels ($\alpha = 0.002$). Life-history traits in this study include length at maturation (*a*), age at maturation (*b*), length at the end of the experiment (*c*), mean length of clutches (*d*), mean clutch size (*e*) and the number of clutches (*f*).

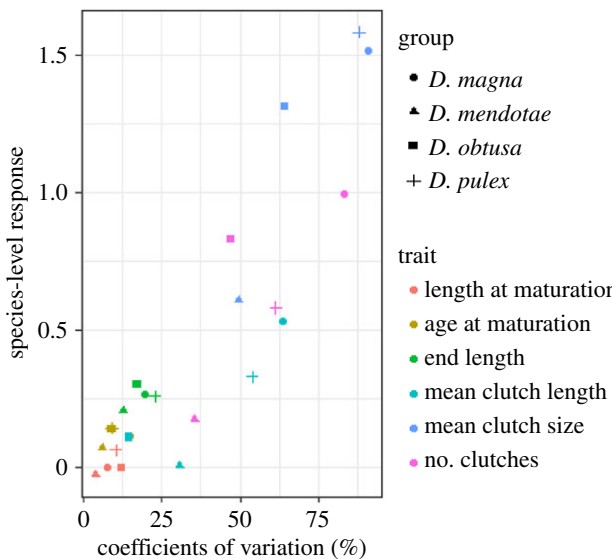

**Figure 2.** The relationship between intraspecific trait variation (COVs) and species-level responses to food quality (Pearson correlation, $r^2 = 0.89$, $t = 9.60$, d.f. $= 22$, $p < 0.01$).

the variance (table 2). Food quality affected all species' current reproduction via clutch size, while it only had significant effects on mean clutch length, which could have implications for fitness of future generations, for larger-bodied *D. magna* and *D. pulex*. Food quality is known to play an important role in growth rate, many studies have used the growth rate hypothesis to predict relationships between growth and P-limited diets or environments (e.g. [31,34]). P-limitation in particular has been well

studied in *Daphnia*; P-intensive processes including organismal growth, rRNA levels and therefore protein production require high availability of phosphorus [21].

Depending on intra- and inter-specific pressures, evolution will favour more or less specialized individuals within a generalist population [52]. Within these populations, clonal variation could lead to a wider environmental range in which the population can maintain fitness under environmental change [53]. Clonal lineages did not have a large contribution to life-history traits, as predicted. This could be potentially due to the relatively small number of clones per species. In addition, these experimental clonal assemblages are somewhat of an artificial construct and raised in the laboratory, which may not reflect field populations. It is enticing that species identity clustered strongly in terms of composite life-history traits (i.e. PC axes) under high-quality (high P) food conditions, but there was no strong species-specific clustering among three of the species under poor nutrient conditions (electronic supplementary material, S2).

## 4.2. Trait variation in body size is constrained, while there is flexibility in reproductive traits

The food quality levels from this study did not support the notion that smaller-bodied *Daphnia* respond more to changes in food quality than larger-bodied *Daphnia*. However, larger *Daphnia* did show more variation in life-history traits. Body size traits were associated with lower amounts of intraspecific variation and species-level responses to food quality, while other reproductive traits were associated with higher amounts of variation and responses to food quality (figure 2). This indicates that *Daphnia* have size-based phenotypes that are somewhat genetically constrained. Allometric constraints may be one possible explanation for conserved morphological traits. It has been shown that regardless of body size, daphniids all follow a similar pattern of resource allocation to growth and reproduction under different levels of food (carbon) quantity [54].

Body size has been implicated in determining sensitivity to food quality, with larger individuals being affected by low food quality more so than smaller individuals [55]. However, there was no evidence in this study that the body size of the species played a role in species-level responses or the amount of intraspecific variation. Previous work has indicated that *Daphnia* may face a competitive trade-off between maximizing growth when food quality is high and minimizing negative effects of poor food quality [31,50]. However, stoichiometric flexibility may allow for changing the C : P ratio of new growth under P-limited conditions in order to avoid consequences of P-limited diets. *D. mendotae* is relatively inflexible compared to the other species in this study in its response to changes in the C : P of its diet, showing strong homeostasis between diets [31]. This matches with this study's results, where *D. mendotae* had overall less intraspecific variation across traits and smaller species-level responses (figure 2).

I hypothesized a trade-off between intraspecific trait variation and species-level responses to food stress. However, in this present study, results were contradictory to expectations: species showed more change between food quality environments with increasing trait variation. This is most likely due to reproductive traits being very P-intensive and very responsive to changes in food quality. In *Daphnia*, P and reproductive trait relationships have not been as well studied as somatic growth rate (SGR), a well-known proxy of fitness [56]. However, shifts toward lower reproduction have been seen for low levels of nitrogen and phosphorus [57] and for low food concentrations [20]; but under toxin-enriched environments, *Daphnia* maintain reproductive output [58]. Plasticity in reproductive traits is generally considered less important in changing population growth rates based on previous modelling of growth and reproductive schedules [30]. These results suggest that potentially environmental buffering from P-limitation has canalized the highly vital growth traits over time, while leaving reproductive rates variable and responsive to environmental change. However, another potential explanation could be that a trade-off would not be expected in this case because these lineages have been raised in the laboratory for many generations with bountiful resources, thereby creating a so-called 'superflea' [59]. This study should be repeated using individuals taken from natural habitats.

# 5. Conclusion

This present study provides evidence that species identity is important in determining life-history traits, but that may not translate into size-structured populations due to variation in reproductive traits across environments that vary in overall food quality. In particular, the flexibility in reproductive traits may play an important role for population persistence in the face of environmental change. Phenotypic plasticity is

the ability of an organism to change its phenotype in response to environmental change. *Daphnia* have shown a great capacity for phenotypic plasticity in predator-avoidance (e.g. [18,19]), nutrient uptake/use efficiency (e.g. [60]) and other life-history traits (e.g. [48]). This study, where a changing environment may select for more responsiveness in reproductive traits, indicates more consideration of evolution of phenotypic plasticity and population persistence through life-history traits [61]. Gathering information about the potential for phenotypically plastic traits via trait variation has been, and will continue to be, a goal toward predicting a species' ability to respond to continued environmental stress. However, there are costs and limits involved in maintaining plastic traits, including genetic and/or developmental constraints, competitive exclusion by a more optimal (and less plastic) trait during a stable period, or geographical limits [62,63]. Species that are flexible in their use of phosphorus may compensate for P-limitation by being more plastic in reproductive life-history traits.

Data accessibility. Data available from the Dryad Digital Repository at: https://doi.org/10.5061/dryad.d4v7g74 [64]. Additional data on effect sizes and data visualizations are available within the electronic supplementary material.
Competing interests. The author declares that she has no competing interests.
Funding. This work was supported by a US National Science Foundation Graduate Research Fellowship Program under grant no. 2013151892.
Acknowledgements. I thank my graduate advisor L.J. Weider, for doing his job extremely well. Special thanks to M. Pfrender, L. Orsini and K. Spitze for providing clonal lineages for *D. magna*, *D. pulex* and *D. obtusa*. I deeply appreciate the input from L.J. Weider, K. Spitze, J. Dudycha, five anonymous reviewers and J. Beyer, whose comments improved this manuscript immensely. Also, I would like to thank K. Roeder, B. Tweedy, E. Freitas and E. Kiehnau for reviewing the earlier drafts of the manuscript. This manuscript constitutes a portion of R. Hartnett's dissertation in partial fulfilment of the PhD requirements at the U. of Oklahoma.

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
