## [Reviewer comments · Royal Society Open Science]

Review History

RSOS-180003.R0 (Original submission)

Review form: Reviewer 1

Is the manuscript scientifically sound in its present form?

No

Are the interpretations and conclusions justified by the results?

No

Is the language acceptable?

No

Is it clear how to access all supporting data?

Yes

Do you have any ethical concerns with this paper?

No

Have you any concerns about statistical analyses in this paper?

Yes

Recommendation?

Reject

Comments to the Author(s)

General comments:

This project investigates the clonal- and species-level variation in life history traits in response to changing nutrient levels. While the methods are sound, the statistical analyses and interpretations are fraught. First, some major conclusions rest on simply visual interpretation of PC plots, rather than rigorous statistical tests (though MANOVAs were used some). Second, definitions of plasticity (in terms of the methods used to quantify plasticity) miss the mark. Using centroid values from PC plots to quantify plasticity (while ignoring the variance within groups in PC space) is likely to misrepresent the degree of plasticity. Furthermore (and perhaps most importantly), there seems to be a confounding of 'sensitivity' and 'plasticity' throughout the manuscript (see multiple comments below). I suggest a very thorough consideration of how to calculate plasticity (e.g., reaction norm approach), and a more careful conveyance of the precise definitions of (adaptive or not) plasticity and sensitivity in the MS.

Line-by-line comments:

Abstract, pg. 2, lines 7-8: While I agree that there are often quite strong links between life history traits and fitness, this is not a quality of life history traits alone. Since this statement is the crux of 'selling' the importance of looking at life history traits, per se, I suggest reframing this.

Pg. 2, line 18: Edit this line to read '...among species across two different environmental conditions...'

Pg. 2, line 25: See comments below regarding how you define plasticity (and thus formulate your predictions)

Pg 3, line 20: Delete 'before death'

Pg 3, line 30: Change 'lowering' to 'decreasing'

Pg 4, lines 25-27: Use 'e.g.,' before these references since many references exist for each of these phenomena

Pg 4: lines 27-33: This sentence is cumbersome and confusing. Please rephrase.

Pg 5, line 22: Change 'are' to 'is' and add 'each of' after 'from'

Pg 7, lines 3-4: Please expand on the 'confounding issue' (do you mean to say that the confounding issue is due to inbreeding?) with clones MA1 and MA2.

Pg 7, line 8: Please report the densities in the housing containers. About how many animals were kept in the 900mL jars and 60mL jars?

Pg 7, line 21: Please provide the methods (could just be a sentence) for determining the amount of

algae that were provided to the maternal lines. Also, some of this information is redundant with the next paragraph – please rectify.

Pg 7, line 49-54: Please provide more detail on how the LoP and HiP treatments were prepared (e.g., more detail than just the C:P ratios).

Pg 8, line 46: Change ‘...to look at the significance of’ to ‘examine the effects of’

Pg 8, line 51: Did you use non-parametric statistics to look at these outcome variables then? Without a statistical test for these variables or a satisfactory explanation for the omission of these tests, these analyses are incomplete.

Pg 9, line 5: ‘Visual inspection of a stem and leaf plot’ is not sufficient detail for how outliers were identified. What are the criteria for the stem and leaf plots used to determine outliers? Usually, it’s that outliers are two SDs from the mean value.

Pg 9, line 37: Change ‘within’ to ‘in’

Pg 9, line 25: Change ‘looked at’ to ‘examined’ or something similar. Check throughout the manuscript that this sort of casual language is corrected (e.g., line 29: ‘In order to look at variance, we looked at...’).

Pg 10, line 5-8: What you call P-sensitivity is essentially a reaction norm (i.e., trait change across phosphorus treatments). This, then, could very reasonably be called plasticity as well. In fact, in many cases, there is more grounding to call this plasticity than the COV and centroid differences. This introduces a crucial area of potential misunderstanding in this manuscript that, indeed, underlies the entire work (even the title!).

Pg 10, line 28-30: I don’t think the inclusion of PC analyses in this paper adds significantly. In fact, it’s quite redundant with the MANOVA results. Visual inspection of a PC plot is enough to give you some idea that groups (here, species) differ in the variables that you’re interested in. But, I would like to see more rigorous statistical investigation of the questions at play. Visual inspection of the PC plot, for example shows separation along PC1 (size variables) for *D. magna* and along PC2 (reproductive variables) for *pulex/obstusa*. But what of statistical significance? That’s why you’ve done the MANOVA.

The other thing that the PC analyses give you (that the MANOVA doesn’t) is the ability to calculate centroids for groups. I have serious concerns, however, about using centroid separation as a measure of plasticity. Centroids, much like point estimates, don’t capture the variance within groups. For example, two groups might have separated centroids (so, by your methods, you’d conclude plasticity), but may, in fact, overlap quite heavily in the (e.g.,) minimum convex polygon that surrounds each group in PC space (due to shifting of the density of points).

Pg 11, line 36: delete the parenthetical ‘(i.e., were more constrained)’. Make sure that when you talk about there being less variance elsewhere in the paper that you don’t jump to the conclusion that phenotypes are more constrained. ‘Constrained’ represents a very specific biological statement about the limits of plasticity that may or may not apply in this case.

Pg 13, line 12: Just to point this out... regarding a previous comment re: constraints, here, the discussion of constraints is merited (that starts with ‘Allometric constraints...’) since it’s discussion/speculation and appropriate language (e.g., ‘may’) is used.

Pg 14, Line 25-28: The distinction between sensitivity and trait variation are hard to understand

throughout this paper and thus need to be better defined. On one hand, if sensitivity is measured as changes in traits from low- to high-P environments (as the equations in the methods define it), then one would expect that, by definition, sensitivity is trait variation (across environments), at least at the clonal level. If the author means to say that sensitivity (at the clonal level) is one thing and trait variation (at the species level) is another, then they should be careful to make this distinction (and remind the reader of this distinction) throughout the MS. Again, since this distinction seems central to the message of the MS, much more careful effort towards both defining and quantifying these must be taken.

Review form: Reviewer 2

Is the manuscript scientifically sound in its present form?

No

Are the interpretations and conclusions justified by the results?

No

Is the language acceptable?

Yes

Is it clear how to access all supporting data?

No

Do you have any ethical concerns with this paper?

No

Have you any concerns about statistical analyses in this paper?

Yes

Recommendation?

Major revision is needed (please make suggestions in comments)

Comments to the Author(s)

Please see comments on the attached document (Appendix A).

Decision letter (RSOS-180003.R0)

30-Nov-2018

Dear Dr Hartnett:

Manuscript ID RSOS-180003 entitled "PLASTICITY AND SENSITIVITY IN LIFE-HISTORY TRAITS AMONG *DAPHNIA* SPECIES UNDER FOOD STRESS" which you submitted to Royal Society Open Science, has been reviewed. The comments from reviewers are included at the bottom of this letter.

In view of the criticisms of the reviewers, the manuscript has been rejected in its current form. However, a new manuscript may be submitted which takes into consideration these comments.

Please note that resubmitting your manuscript does not guarantee eventual acceptance, and that your resubmission will be subject to peer review before a decision is made.

Your resubmitted manuscript should be submitted by 30-May-2019. If you are unable to submit by this date please contact the Editorial Office.

Please note that Royal Society Open Science will introduce article processing charges for all new submissions received from 1 January 2018. Charges will also apply to papers transferred to Royal Society Open Science from other Royal Society Publishing journals, as well as papers submitted as part of our collaboration with the Royal Society of Chemistry (<http://rsos.royalsocietypublishing.org/chemistry>). If your manuscript is submitted and accepted for publication after 1 Jan 2018, you will be asked to pay the article processing charge, unless you request a waiver and this is approved by Royal Society Publishing. You can find out more about the charges at <http://rsos.royalsocietypublishing.org/page/charges>. Should you have any queries, please contact openscience@royalsociety.org.

on behalf of Dr Michael Tobler (Associate Editor) and Professor Kevin Padian (Subject Editor)
openscience@royalsociety.org

Associate Editor Comments to Author (Dr Michael Tobler):

We have received the feedback from two reviewers. Both see merit in the study and agreed that the experimental approach was sound, but they also found significant issues particularly associated with the statistical analyses. Accordingly, the manuscript will need significant revision and cannot be accepted for publication in its present form. However, if the author can address the reviewers' concerns and revise the manuscript accordingly, the paper should be suitable for publication in RSOS.

Reviewers' Comments to Author:

Reviewer: 1

Comments to the Author(s)

General comments:

This project investigates the clonal- and species-level variation in life history traits in response to

changing nutrient levels. While the methods are sound, the statistical analyses and interpretations are fraught. First, some major conclusions rest on simply visual interpretation of PC plots, rather than rigorous statistical tests (though MANOVAs were used some). Second, definitions of plasticity (in terms of the methods used to quantify plasticity) miss the mark. Using centroid values from PC plots to quantify plasticity (while ignoring the variance within groups in PC space) is likely to misrepresent the degree of plasticity. Furthermore (and perhaps most importantly), there seems to be a confounding of 'sensitivity' and 'plasticity' throughout the manuscript (see multiple comments below). I suggest a very thorough consideration of how to calculate plasticity (e.g., reaction norm approach), and a more careful conveyance of the precise definitions of (adaptive or not) plasticity and sensitivity in the MS.

Line-by-line comments:

Abstract, pg. 2, lines 7-8: While I agree that there are often quite strong links between life history traits and fitness, this is not a quality of life history traits alone. Since this statement is the crux of 'selling' the importance of looking at life history traits, per se, I suggest reframing this.

Pg. 2, line 18: Edit this line to read '...among species across two different environmental conditions...'

Pg. 2, line 25: See comments below regarding how you define plasticity (and thus formulate your predictions)

Pg 3, line 20: Delete 'before death'

Pg 3, line 30: Change 'lowering' to 'decreasing'

Pg 4, lines 25-27: Use 'e.g.,' before these references since many references exist for each of these phenomena

Pg 4: lines 27-33: This sentence is cumbersome and confusing. Please rephrase.

Pg 5, line 22: Change 'are' to 'is' and add 'each of' after 'from'

Pg 7, lines 3-4: Please expand on the 'confounding issue' (do you mean to say that the confounding issue is due to inbreeding?) with clones MA1 and MA2.

Pg 7, line 8: Please report the densities in the housing containers. About how many animals were kept in the 900mL jars and 60mL jars?

Pg 7, line 21: Please provide the methods (could just be a sentence) for determining the amount of algae that were provided to the maternal lines. Also, some of this information is redundant with the next paragraph - please rectify.

Pg 7, line 49-54: Please provide more detail on how the LoP and HiP treatments were prepared (e.g., more detail than just the C:P ratios).

Pg 8, line 46: Change '...to look at the significance of' to 'examine the effects of'

Pg 8, line 51: Did you use non-parametric statistics to look at these outcome variables then? Without a statistical test for these variables or a satisfactory explanation for the omission of these tests, these analyses are incomplete.

Pg 9, line 5: 'Visual inspection of a stem and leaf plot' is not sufficient detail for how outliers were identified. What are the criteria for the stem and leaf plots used to determine outliers? Usually, it's that outliers are two SDs from the mean value.

Pg 9, line 37: Change 'within' to 'in'

Pg 9, line 25: Change 'looked at' to 'examined' or something similar. Check throughout the manuscript that this sort of casual language is corrected (e.g., line 29: 'In order to look at variance, we looked at...').

Pg 10, line 5-8: What you call P-sensitivity is essentially a reaction norm (i.e., trait change across phosphorus treatments). This, then, could very reasonably be called plasticity as well. In fact, in many cases, there is more grounding to call this plasticity than the COV and centroid differences. This introduces a crucial area of potential misunderstanding in this manuscript that, indeed, underlies the entire work (even the title!).

Pg 10, line 28-30: I don't think the inclusion of PC analyses in this paper adds significantly. In fact, it's quite redundant with the MANOVA results. Visual inspection of a PC plot is enough to give you some idea that groups (here, species) differ in the variables that you're interested in. But, I would like to see more rigorous statistical investigation of the questions at play. Visual inspection of the PC plot, for example shows separation along PC1 (size variables) for *D. magna* and along PC2 (reproductive variables) for *pulex/obstusa*. But what of statistical significance? That's why you've done the MANOVA.

The other thing that the PC analyses give you (that the MANOVA doesn't) is the ability to calculate centroids for groups. I have serious concerns, however, about using centroid separation as a measure of plasticity. Centroids, much like point estimates, don't capture the variance within groups. For example, two groups might have separated centroids (so, by your methods, you'd conclude plasticity), but may, in fact, overlap quite heavily in the (e.g.) minimum convex polygon that surrounds each group in PC space (due to shifting of the density of points).

Pg 11, line 36: delete the parenthetical '(i.e., were more constrained)'. Make sure that when you talk about there being less variance elsewhere in the paper that you don't jump to the conclusion that phenotypes are more constrained. 'Constrained' represents a very specific biological statement about the limits of plasticity that may or may not apply in this case.

Pg 13, line 12: Just to point this out... regarding a previous comment re: constraints, here, the discussion of constraints is merited (that starts with 'Allometric constraints...') since it's discussion/speculation and appropriate language (e.g., 'may') is used.

Pg 14, Line 25-28: The distinction between sensitivity and trait variation are hard to understand throughout this paper and thus need to be better defined. On one hand, if sensitivity is measured as changes in traits from low- to high-P environments (as the equations in the methods define it), then one would expect that, by definition, sensitivity is trait variation (across environments), at least at the clonal level. If the author means to say that sensitivity (at the clonal level) is one thing and trait variation (at the species level) is another, then they should be careful to make this distinction (and remind the reader of this distinction) throughout the MS. Again, since this distinction seems central to the message of the MS, much more careful effort towards both defining and quantifying these must be taken.

Reviewer: 2

Comments to the Author(s)

Please see comments on the attached document.

Author's Response to Decision Letter for (RSOS-180003.R0)

See Appendix B.

RSOS-191024.R0

Review form: Reviewer 1

Is the manuscript scientifically sound in its present form?

Yes

Are the interpretations and conclusions justified by the results?

Yes

Is the language acceptable?

Yes

Do you have any ethical concerns with this paper?

No

Recommendation?

Accept with minor revision (please list in comments)

Comments to the Author(s)

The authors present a revised version of their manuscript in which they explore interspecific variation in life history traits in response to nutritional environments. They have done well to eliminate the PC analyses from the manuscript and focus instead on MANOVA analyses, which strengthen the overall statistical approach. The findings are generally expected, but interesting. The overall comparative approach is a strength.

Most of my critiques at this point are minor (see below). I would, however, like to see some effort spent in clarifying the expectation that mean and variance in responses should be inverse correlated. Indeed, my expectation for many biological traits is the opposite: that mean and variance in responses are generally positively correlated. This ends up bearing somewhat heavily on the manuscript since this prediction is introduced in the introduction and its implications are discussed in the discussion.

Line-by-line edits

Pg 5, Line 3: delete 'large'

Pg 5, Lines 31-34: Alternatively, mean and variance is often positively correlated in biological data. Part of this is a 'floor effect', for example. I would suggest discussing the alternative hypothesis (that mean and variance are positively correlated) here.

Pg 6, Line 41: add 'pond' (?) after 'ephemeral'

Pg 7, Line 26: what volume of algae (at 1mg C L^{-1}) was fed to the zooplankton?

Pg 9, Lines 3-4: Missing word(s) here?

Pg 9, Line 26: change 'was run' to 'was conducted' – also change this in future sentences, only because 'was run' sounds a bit awkward

Pg 10, Line 54: Please make explicit the direction of effect here – larger species have greater degrees of variation?

Pg 11, Lines 36-38: I suggest changing this heading so that it reads '[...] and species identity explained greater trait variation than genotype', or something along those lines.

Pg 11, Line 43: Change 'imposes' to 'is imposed'

Pg 14, Line 12: Missing date for reference

Figure 1: When abbreviating for microns, use mu instead of the letter 'u', in y-axes labels

Figure 2: I suggest having a white background for this figure given that some of the points are quite light gray.

Review form: Reviewer 3

Is the manuscript scientifically sound in its present form?

Yes

Are the interpretations and conclusions justified by the results?

Yes

Is the language acceptable?

Yes

Do you have any ethical concerns with this paper?

No

Recommendation?

Accept with minor revision (please list in comments)

Comments to the Author(s)

I have a few minor comments that should be addressed for more transparency.

Introduction:

P5 L16 "therefore clonal variation may significantly add to genetic contributions". The meaning

of this sentence is cryptic, can you try to reformulate?

Methods

P6 L40 I think one word is missing in the sentence “finnish pond from a ephemeral with dessication”

“incidental ambient lighting”: I suppose not only the stock culture but also the experiment were in these conditions. When was the experiment conducted?

Please provide more details about spectrophotometer measurements: wavelength, and perhaps a reference to the literature on this already established procedure?

Please provide details about the experimental animals. Especially in smaller species, a single clutch contains less than 20 neonates, so to reach the indicated number clutches must have been pooled. Were all experimental animals born on the same day from different mothers, or were they all from the same mother but born on different days? I am aware this is a logistical issue but also important when interpreting results, because it influences the observed variance.

P9 word missing “mortality accounts for less than five percent...”

P13 (middle) replace “we hypothesize” with “I hypothesize”

P14 The superflea Tessier reference is incomplete, I suppose it was a formatting error.

There is no R code available in supp Mat, please provide it

Figure 2 is hard to read. Please consider using a white background, as grey on grey is not easy. Is RSOS having a limitation on color figures? If not I would consider using colors in this figure, because even if “only” 4 shades of grey aren’t much, it is a bit hard. I also wish the symbols were larger, and I am wondering why the symbols for mean clutch size look different in shape?

Decision letter (RSOS-191024.R0)

03-Jul-2019

Dear Dr Hartnett

On behalf of the Editor, I am pleased to inform you that your Manuscript RSOS-191024 entitled "VARIATION IN LIFE-HISTORY TRAITS AMONG *DAPHNIA* AND ITS RELATIONSHIP TO SPECIES-LEVEL RESPONSES TO PHOSPHORUS LIMITATION" has been accepted for publication in Royal Society Open Science subject to minor revision in accordance with the referee suggestions. Please find the referees' comments at the end of this email.

The reviewers and Subject Editor have recommended publication, but also suggest some minor revisions to your manuscript. Therefore, I invite you to respond to the comments and revise your manuscript.

- Ethics statement

- Data accessibility

It is a condition of publication that all supporting data are made available either as supplementary information or preferably in a suitable permanent repository. The data accessibility section should state where the article's supporting data can be accessed. This section should also include details, where possible of where to access other relevant research materials such as statistical tools, protocols, software etc can be accessed. If the data has been deposited in

an external repository this section should list the database, accession number and link to the DOI for all data from the article that has been made publicly available. Data sets that have been deposited in an external repository and have a DOI should also be appropriately cited in the manuscript and included in the reference list.

<http://datadryad.org/submit?journalID=RSOS&manu=RSOS-191024>

- **Competing interests**

- **Authors' contributions**

- **Acknowledgements**

- **Funding statement**

Because the schedule for publication is very tight, it is a condition of publication that you submit the revised version of your manuscript before 12-Jul-2019. Please note that the revision deadline will expire at 00.00am on this date. If you do not think you will be able to meet this date please let me know immediately.

on behalf of Dr Michael Tobler (Associate Editor) and Kevin Padian (Subject Editor)
openscience@royalsociety.org

Associate Editor Comments to Author (Dr Michael Tobler):

The manuscript was re-assessed by two reviewers and both generally agree that the author has adequately revised the manuscript. They provide some additional feedback that should help to improve the manuscript. I would particularly like to echo two reviewer comments, one pertaining the clarification of why a negative correlation between mean and variance are expected, and one pertaining the removal of gray background in the figures. In the context of the

latter, I should add that color figures are free, and using color may help to make figure 2 more clear.

Reviewer comments to Author:

Reviewer: 1

Comments to the Author(s)

The authors present a revised version of their manuscript in which they explore interspecific variation in life history traits in response to nutritional environments. They have done well to eliminate the PC analyses from the manuscript and focus instead on MANOVA analyses, which strengthen the overall statistical approach. The findings are generally expected, but interesting. The overall comparative approach is a strength.

Most of my critiques at this point are minor (see below). I would, however, like to see some effort spent in clarifying the expectation that mean and variance in responses should be inverse correlated. Indeed, my expectation for many biological traits is the opposite: that mean and variance in responses are generally positively correlated. This ends up bearing somewhat heavily on the manuscript since this prediction is introduced in the introduction and its implications are discussed in the discussion.

Line-by-line edits

Pg 5, Line 3: delete 'large'

Pg 5, Lines 31-34: Alternatively, mean and variance is often positively correlated in biological data. Part of this is a 'floor effect', for example. I would suggest discussing the alternative hypothesis (that mean and variance are positively correlated) here.

Pg 6, Line 41: add 'pond' (?) after 'ephemeral'

Pg 7, Line 26: what volume of algae (at 1mg C L^{-1}) was fed to the zooplankton?

Pg 9, Lines 3-4: Missing word(s) here?

Pg 9, Line 26: change 'was run' to 'was conducted' - also change this in future sentences, only because 'was run' sounds a bit awkward

Pg 10, Line 54: Please make explicit the direction of effect here - larger species have greater degrees of variation?

Pg 11, Lines 36-38: I suggest changing this heading so that it reads '[...] and species identity explained greater trait variation than genotype', or something along those lines.

Pg 11, Line 43: Change 'imposes' to 'is imposed'

Pg 14, Line 12: Missing date for reference

Figure 1: When abbreviating for microns, use mu instead of the letter 'u', in y-axis labels

Figure 2: I suggest having a white background for this figure given that some of the points are quite light gray.

Reviewer: 3

Comments to the Author(s)

I have a few minor comments that should be addressed for more transparency.

Introduction:

P5 L16 "therefore clonal variation may significantly add to genetic contributions". The meaning of this sentence is cryptic, can you try to reformulate?

Methods

P6 L40 I think one word is missing in the sentence "finnish pond from a ephemeral with dessication"

"incidental ambient lighting": I suppose not only the stock culture but also the experiment were in these conditions. When was the experiment conducted?

Please provide more details about spectrophotometer measurements: wavelength, and perhaps a reference to the literature on this already established procedure?

Please provide details about the experimental animals. Especially in smaller species, a single clutch contains less than 20 neonates, so to reach the indicated number clutches must have been pooled. Were all experimental animals born on the same day from different mothers, or were they all from the same mother but born on different days? I am aware this is a logistical issue but also important when interpreting results, because it influences the observed variance.

P9 word missing "mortality accounts for less than five percent..."

P13 (middle) replace "we hypothesize" with "I hypothesize"

P14 The superflea Tessier reference is incomplete, I suppose it was a formatting error.

There is no R code available in supp Mat, please provide it

Figure 2 is hard to read. Please consider using a white background, as grey on grey is not easy. Is RSOS having a limitation on color figures? If not I would consider using colors in this figure, because even if "only" 4 shades of grey aren't much, it is a bit hard. I also wish the symbols were larger, and I am wondering why the symbols for mean clutch size look different in shape?

Author's Response to Decision Letter for (RSOS-191024.R0)

See Appendices C & D.

Decision letter (RSOS-191024.R1)

17-Jul-2019

Dear Dr Hartnett,

I am pleased to inform you that your manuscript entitled "VARIATION IN LIFE-HISTORY TRAITS AMONG *DAPHNIA* AND ITS RELATIONSHIP TO SPECIES-LEVEL RESPONSES TO PHOSPHORUS LIMITATION" is now accepted for publication in Royal Society Open Science.

You can expect to receive a proof of your article in the near future. Please contact the editorial office (openscience_proofs@royalsociety.org and openscience@royalsociety.org) to let us know if

you are likely to be away from e-mail contact. Due to rapid publication and an extremely tight schedule, if comments are not received, your paper may experience a delay in publication.

on behalf of Dr Michael Tobler (Associate Editor) and Kevin Padian (Subject Editor)
openscience@royalsociety.org

Appendix A

This study explores life-history trait variation within/among species and proposes to examine the effects of genetics/environment, body size, and plasticity/sensitivity on daphnid trait expression. Overall, the experimental methods seem to be mostly solid, and the importance of variation in reproductive traits with nutrient limitation is a key finding that has been relatively overlooked in the stoichiometric literature. This study could make an important scientific contribution; however, I found myself getting lost at certain points and questioning the rational and interpretation behind some of the predictions and results. Therefore, I think that the manuscript might benefit by addressing some major and minor points.

Major Points

Structure

1. The study questions are clearly laid out at the end of the introduction, which I appreciate. But, I believe that they can be better setup in the introduction and discussed in the later parts of the manuscript. There are introductory paragraphs about food quality, clones and plasticity, which sets up 2 of the 3 study questions, but nothing about body size in the introduction. Please consider adding a body size paragraph to justify rational for why organisms of different body size might be expected to grow differently. Also, while you do talk about plasticity you don't say how it might be related to sensitivity until the final intro paragraph, which left me thoroughly confused. You hint at a mechanism in the discussion, but it would be easier for the reader if you lay this out in the plasticity paragraph.

2. I might've missed it, but I only found one mention of the first study aim in the results and not really much discussion of it in the discussion section. This might make for an interesting discussion paragraph along with talkin about the the interactions between genotype and the environment. These are also tested but not thoroughly discussed in the paper.

Stats

3. There was some death in these experiments, but as survival data was not presented, I don't have a feel for how this might've influenced your results. I understand if survival wasn't part of your story, but please report the proportion of individuals that died for each species so that we may assess the robustness of the results. Similarly, please explain the rational for the 5 day survival cutoff reported on Page 7 Line 25? It seems to me that there would still be a significant different between body sizes of day 7 and day 20 animals which could affect your variance estimates.

4. As the PCA is central to your story, the reader could use some extra information to help interpret it. Please state whether your variables were centered and scaled so that we can be certain that individual variables didn't have a greater influence on the ordination. Also, you report in the MANOVA calculation that certain variables were skewed. How was this handled in the PCA ordination? Finally, please report the correlations between each variable and axis so that we can confirm that each PC axis was related to "growth" and reproduction as you say.

5. I'm still really confused about how the COV-sensitivity analysis was conducted. The COV's seem to be calculated with diet treatments for each species and for each individual trait whereas

the sensitivity analysis is calculated on multivariate data for each species. My question is if you are including both food treatment and trait variation (individual on the x-axis and multivariate on the y) on both axes, can we actually consider these two measurements to be independent? To me it seems self-evident that univariate and multivariate trait variation should be positively correlated. Please provide further description and justification for this analysis.

Results Interpretation

6. Since growth wasn't calculated using the common metric mass specific growth rate, I'm not sure that growth is the most accurate way to describe this axis. Specifically, since you don't take into account different starting masses, there's no way to directly compare growth rates across species. SAM is also generally considered to be a reproductively related trait as it is assumed to be under selection from size-selective predators and is inherently related to reproduction. I would be more comfortable if you referred to this axis as relating to body size.

7. I found myself questioning some of the results interpretation in the discussion section.

A) In the first two paragraphs, you talk about the buffering effects of clonal variation and how this might help species persistence in different or variable environments. However, phenotypic variation among clones was the weakest of any variable in your study and as you say these clones weren't from the same populations, so it's not clear to me how genotypic variation that does not lead to phenotypic differences among clones might be adaptive in natural environments.

B) In the 3rd paragraph, I'm not sure that your results show that larger-bodied daphnia are more affected by food quality than smaller bodied daphnia. Table 3 shows that this pattern is trait specific, Figure 2 shows that *D. obtusa* appears to be more variable than *D. magna* and figure 4 seems to show that *obtusa* is the most variable of the 3 species.

Minor Points

P3 L31: It is debatable whether P is the *ultimate* limiting nutrient in freshwater systems. This supposition is not supported by your reference to Sterner 2008 and certainly not by the meta analyses of Elser et al. 2007 or any of the multitude of studies examining N & P co-limitation conducted in the last decade or so. It certainly is an important limiting element, but not necessarily the *most* important.

P3 L30: It's true that daphnids could be more susceptible to P-limitation, but as most environments are experiencing P-loading rather than P reductions it is unclear to me how they might be used as an indicator organism. Wouldn't it make more sense to use an organism sensitive to excess P?

P4 L12: "Clonal variation can be considered as important as species identity" Please explain what this importance is referring to.

P5 L5: Are these predictions here? Do they all refer to the final study question 3? If not, you might consider moving them up with their corresponding question or spelling out some of these connections specifically in respect to question 3 earlier in the paragraph bc this information could be confusing out of context.

P6 L3: Another caveat that you might want to add is that these animals have been adapted to lab conditions for long periods of time which might influence the magnitude of their plastic responses compared to animals in the wild.

P6 L47: So the total amount fed to these animals would be 0.5 mg C day? Can you show that this level of food is not limiting to the larger *magna* and *pulex* species given that there were also likely neonates feeding in these tubes later on in the experiment? Could this account for some of the trait variance in these larger body sized animals?

P7 L38-45: These sentences might be better off in the results section.

P9 L12: Do you have quantitative statistics to support this? I only see descriptive statistics for individual traits in the tables and figures.

P9 L38: This shift might be moved to the discussion where it can be discussed further.

P12 L34: It's not clear how these phylogenetic differences could've been related to plasticity. The work by Seidendorf 2010 doesn't support this, but if you believe that this might be the case, would it be worth including it as a covariate in your analyses?

P12 L51: You might consider changing this to "partially matches the prediction". There might be an individual *magna* and *pulex* clone that is more sensitive, but on average *obtusa* seems to be the most sensitive species.

P13 L19-21: It would be really helpful to develop this idea in the introduction.

P13 L38: Hood and Sterner 2014 didn't measure clutch frequency or size.

P13 L 43: I really like this idea, so if you get a chance you could expand upon it in future versions of this manuscript or if not in future work.

Tables and Figures: It might help if you rearrange these to put large- and small-bodied animals together.

The legends for Figure 3 could also do with a bit of reformatting.

Appendix B

General comments to the editor and reviewers:

I very much appreciated the care and thought put into the reviews of this manuscript. Both reviewers had major concerns over the robustness of my analysis and therefore the conclusions/interpretations that I drew from the study. Therefore, I have completely redone my analyses and took special care to not draw any conclusions/discussion points qualitatively. I have relegated the PC analysis, used for visual interpretation, to the supplemental materials, and focused on the MANOVA for interpretation as suggested by reviewer 1. In addition, I ran two-sample t-tests to quantitatively test the claims I made on daphniid size and their effects on responsiveness and variation (before sensitivity and plasticity, see next paragraph for changes in terms). I also ran a correlation analysis to determine quantitatively the association between species-level responses and intraspecific variation. Because some of these new analyses changed my results, the discussion has been substantially changed.

In addition, another major concern was the way that I had (mis)defined plasticity and sensitivity. I have clarified my manuscript to investigate intraspecific trait variation (COVs) and species-level responses (mean trait in HP – mean trait in LP: once defined as sensitivity when it could be plasticity) to remove any misunderstanding of what I am trying to address. This has been reflected throughout the manuscript, and should adequately remove this major concern of confounding sensitivity and plasticity.

Other major revisions:

I revised the introduction to focus less on the stress-aspect of the environmental treatment and more on the responses *Daphnia* exhibit under differing food quality, as well as ensuring that I adequately discuss each of the study's aims.

Effect size calculations, and an additional two-way ANOVA, was moved to the supplementary materials, as they did not add anything to the main message of the manuscript after re-doing the other analyses and the results of the ANOVA were insignificant, so they have been relegated to supplements.

Minor revisions:

I have revised the manuscript with grammatical suggestions by both reviewers. See line-by-line comments/revisions below for more information.

Line by line comments:

Reviewer 1

Comments to the Author(s)

General comments:

This project investigates the clonal- and species-level variation in life history traits in response to changing nutrient levels. While the methods are sound, the statistical analyses and interpretations are fraught. First, some major conclusions rest on simply visual interpretation of PC plots, rather than rigorous statistical tests (though MANOVAs were used some). Second, definitions of plasticity (in terms of the methods used to quantify plasticity) miss the mark. Using centroid values from PC plots to quantify plasticity (while ignoring the variance within groups in PC space) is likely to misrepresent the degree of plasticity. Furthermore (and perhaps most importantly), there seems to be a confounding of 'sensitivity' and 'plasticity' throughout the manuscript (see multiple comments below). I suggest a very thorough consideration of how to calculate plasticity (e.g., reaction norm approach), and a more careful conveyance of the precise definitions of (adaptive or not) plasticity and sensitivity in the MS.

Please see general comments above for a description of how these major concerns were addressed.

Line-by-line comments:

Abstract, pg. 2, lines 7-8: While I agree that there are often quite strong links between life history traits and fitness, this is not a quality of life history traits alone. Since this statement is the crux of 'selling' the importance of looking at life history traits, per se, I suggest reframing this.

Rephrased as suggested.

Pg. 2, line 18: Edit this line to read '...among species across two different environmental conditions...'

Edited as suggested

Pg. 2, line 25: See comments below regarding how you define plasticity (and thus formulate your predictions)

See general comments to the editor. I have re-evaluated how I would like to define terms and make predictions.

Pg 3, line 20: Delete 'before death'

Edited as suggested

Pg 3, line 30: Change 'lowering' to 'decreasing'

Edited as suggested

Pg 4, lines 25-27: Use 'e.g.,' before these references since many references exist for each of these phenomena

Edited as suggested

Pg 4: lines 27-33: This sentence is cumbersome and confusing. Please rephrase.

Edited as suggested.

Pg 5, line 22: Change 'are' to 'is' and add 'each of' after 'from'

Edited as suggested.

Pg 7, lines 3-4: Please expand on the 'confounding issue' (do you mean to say that the confounding issue is due to inbreeding?) with clones MA1 and MA2.

Rephrased to clarify that yes, I did mean that inbreeding could be a potential confounding factor for the amount of variation/response seen.

Pg 7, line 8: Please report the densities in the housing containers. About how many animals were kept in the 900mL jars and 60mL jars?

I rephrased to indicate that maternal lines were raised with only one individual per 60ml container. The best information I have for the housing containers is the minimum and maximum number of adults per jar. So I added an approximate range of densities based on that.

Pg 7, line 21: Please provide the methods (could just be a sentence) for determining the amount of algae that were provided to the maternal lines. Also, some of this information is redundant with the next paragraph – please rectify.

I added a sentence to indicate that we used a spectrophotometer to correlate chlorophyll to carbon. I think I can understand the confusion for the redundancy. I took adults from stock cultures and raised them individually in 60ml jars. These adults produced the experimental animals that I took measurements for. I have done my best to clarify that process.

Pg 7, line 49-54: Please provide more detail on how the LoP and HiP treatments were prepared (e.g., more detail than just the C:P ratios).

Added as suggested.

Pg 8, line 46: Change '...to look at the significance of' to 'examine the effects of'

Edited as suggested.

Pg 8, line 51: Did you use non-parametric statistics to look at these outcome variables then? Without a

statistical test for these variables or a satisfactory explanation for the omission of these tests, these analyses are incomplete.

I had not. I included the variables that were not parametric and checked the post-hoc residuals to make sure that multivariate normality was still met.

Pg 9, line 5: 'Visual inspection of a stem and leaf plot' is not sufficient detail for how outliers were identified. What are the criteria for the stem and leaf plots used to determine outliers? Usually, it's that outliers are two SDs from the mean value.

I changed the criteria for removing outliers to +/- s.d. as suggested.

Pg 9, line 37: Change 'within' to 'in'

Edited as suggested.

Pg 9, line 25: Change 'looked at' to 'examined' or something similar. Check throughout the manuscript that this sort of casual language is corrected (e.g., line 29: 'In order to look at variance, we looked at...').

Edited as suggested.

Pg 10, line 5-8: What you call P-sensitivity is essentially a reaction norm (i.e., trait change across phosphorus treatments). This, then, could very reasonably be called plasticity as well. In fact, in many cases, there is more grounding to call this plasticity than the COV and centroid differences. This introduces a crucial area of potential misunderstanding in this manuscript that, indeed, underlies the entire work (even the title!).

See general comments to the editor. I changed the approach of my analysis to address this comment.

Pg 10, line 28-30: I don't think the inclusion of PC analyses in this paper adds significantly. In fact, it's quite redundant with the MANOVA results. Visual inspection of a PC plot is enough to give you some idea that groups (here, species) differ in the variables that you're interested in. But, I would like to see more rigorous statistical investigation of the questions at play. Visual inspection of the PC plot, for example shows separation along PC1 (size variables) for *D. magna* and along PC2 (reproductive variables) for *pulex/obtusata*. But what of statistical significance? That's why you've done the MANOVA.

The other thing that the PC analyses give you (that the MANOVA doesn't) is the ability to calculate centroids for groups. I have serious concerns, however, about using centroid separation as a measure of plasticity. Centroids, much like point estimates, don't capture the variance within groups. For example, two groups might have separated centroids (so, by your methods, you'd conclude plasticity), but may, in fact, overlap quite heavily in the (e.g.,) minimum convex polygon that surrounds each group in PC space (due to shifting of the density of points).

See general comments to the editor. I changed the approach of my analysis and interpretation to address this comment.

Pg 11, line 36: delete the parenthetical '(i.e., were more constrained)'. Make sure that when you talk about there being less variance elsewhere in the paper that you don't jump to the conclusion that phenotypes are more constrained. 'Constrained' represents a very specific biological statement about the limits of plasticity that may or may not apply in this case.

Edited as suggested.

Pg 13, line 12: Just to point this out... regarding a previous comment re: constraints, here, the discussion of constraints is merited (that starts with 'Allometric constraints...') since it's discussion/speculation and appropriate language (e.g., 'may') is used.

A welcome clarification, but no edit indicated here.

Pg 14, Line 25-28: The distinction between sensitivity and trait variation are hard to understand throughout this paper and thus need to be better defined. On one hand, if sensitivity is measured as changes in traits from low- to high-P environments (as the equations in the methods define it), then one would expect that, by definition, sensitivity is trait variation (across environments), at least at the clonal

level. If the author means to say that sensitivity (at the clonal level) is one thing and trait variation (at the species level) is another, then they should be careful to make this distinction (and remind the reader of this distinction) throughout the MS. Again, since this distinction seems central to the message of the MS, much more careful effort towards both defining and quantifying these must be taken.

See general comments to the editor. I agree that I may not fully understand the distinction between these terms. Having a reader restate my definitions really showed me how unclear they were. I admit that I could not understand how the reviewer was interpreting my terms. I have taken much more care to think about how to define and quantify these calculations. I have decided to reframe the manuscript's terms altogether to avoid misunderstanding. I have classified changes in traits from low to high as species-level responses and made sure my trait variation was calculated at the species level, giving the amount of variation within each species (intraspecific variation) and did not make any claims to compare variation within or between clones.

Reviewer: 2

Comments to the Author(s)

This study explores life-history trait variation within/among species and proposes to examine the effects of genetics/environment, body size, and plasticity/sensitivity on daphnid trait expression. Overall, the experimental methods seem to be mostly solid, and the importance of variation in reproductive traits with nutrient limitation is a key finding that has been relatively overlooked in the stoichiometric literature. This study could make an important scientific contribution; however, I found myself getting lost at certain points and questioning the rationale and interpretation behind some of the predictions and results. Therefore, I think that the manuscript might benefit by addressing some major and minor points.

Major Points

Structure

1. The study questions are clearly laid out at the end of the introduction, which I appreciate. But, I believe that they can be better setup in the introduction and discussed in the later parts of the manuscript. There are introductory paragraphs about food quality, clones and plasticity, which sets up 2 of the 3 study questions, but nothing about body size in the introduction. Please consider adding a body size paragraph to justify rationale for why organisms of different body size might be expected to grow differently. Also, while you do talk about plasticity you don't say how it might be related to sensitivity until the final intro paragraph, which left me thoroughly confused. You hint at a mechanism in the discussion, but it would be easier for the reader if you lay this out in the plasticity paragraph.

A paragraph was added to explain the rationale behind why smaller-bodied Daphnia are expected to respond differently than large-bodied Daphnia.

2. I might've missed it, but I only found one mention of the first study aim in the results and not really much discussion of it in the discussion section. This might make for an interesting discussion paragraph along with talking about the interactions between genotype and the environment. These are also tested but not thoroughly discussed in the paper.

The introduction was re-structured in order to provide more background on the first study aim. In addition, the discussion was expanded to spend more time on the results between genetic and environmental contributions to life-history traits.

Stats

3. There was some death in these experiments, but as survival data was not presented, I don't have a feel for how this might've influenced your results. I understand if survival wasn't part of your story, but please report the proportion of individuals that died for each species so that we may assess the robustness of the results. Similarly, please explain the rationale for the 5 day survival cutoff reported on Page 7 Line 25? It seems to me that there would still be a significant difference between body sizes of day 7 and day 20 animals which could affect your variance estimates.

If the animal did not make it to 5 days, it was considered an unexpected death in the study, and was removed because it would bias the amount of size information available compared to reproductive data (it did not reproduce because it was dead, not because of food stress). Survivorship information has been

added to the manuscript for transparency. While I do not feel like survivorship data needs to be reported directly, as it accounted for less than 5% of the study, I did include a chi-square contingency analysis to ensure that survivorship was independent of species identity and treatment.

4. As the PCA is central to your story, the reader could use some extra information to help interpret it. Please state whether your variables were centered and scaled so that we can be certain that individual variables didn't have a greater influence on the ordination. Also, you report in the MANOVA calculation that certain variables were skewed. How was this handled in the PCA ordination? Finally, please report the correlations between each variable and axis so that we can confirm that each PC axis was related to "growth" and reproduction as you say.

See general comments to the editor. I changed the approach of my analysis as another reviewer pointed out that visual inspection of the PC plots is qualitative assessment rather than qualitative. I have moved the plots to the supplements for visual purposes only. For visual interpretation only, it is acceptable to ignore assumptions of normality for variables (Tabachnick and Fidell). I did add the loadings for the PC analysis.

5. I'm still really confused about how the COV-sensitivity analysis was conducted. The COV's seem to be calculated with diet treatments for each species and for each individual trait whereas the sensitivity analysis is calculated on multivariate data for each species. My question is if you are including both food treatment and trait variation (individual on the x-axis and multivariate on the y) on both axes, can we actually consider these two measurements to be independent? To me it seems self-evident that univariate and multivariate trait variation should be positively correlated. Please provide further description and justification for this analysis.

I corrected this analysis to ensure that COV and sensitivity (now labeled "species-level responses" in the manuscript) were both calculated for each species and for each trait. Therefore the correlation comparison is multivariate overall and post-hoc univariate correlations are also reported.

Results Interpretation

6. Since growth wasn't calculated using the common metric mass specific growth rate, I'm not sure that growth is the most accurate way to describe this axis. Specifically, since you don't take into account different starting masses, there's no way to directly compare growth rates across species. SAM is also generally considered to be a reproductively related trait as it is assumed to be under selection from size-selective predators and is inherently related to reproduction. I would be more comfortable if you referred to this axis as relating to body size.

I agree with this criticism. I have changed the axis to PC and interpretation from growth to size.

7. I found myself questioning some of the results interpretation in the discussion section. A) In the first two paragraphs, you talk about the buffering effects of clonal variation and how this might help species persistence in different or variable environments. However, phenotypic variation among clones was the weakest of any variable in your study and as you say these clones weren't from the same populations, so it's not clear to me how genotypic variation that does not lead to phenotypic differences among clones might be adaptive in natural environments. B) In the 3rd paragraph, I'm not sure that your results show

that larger-bodied daphnia are more affected by food quality than smaller bodied daphnia. Table 3 shows that this pattern is trait specific, Figure 2 shows that *D. obtusa* appears to be more variable than *D. magna* and figure 4 seems to show that *obtusa* is the most variable of the 3 species.

The reviewer is correct on both fronts. Clonal variation and size did not affect responses to food stress like I had claimed. Re-doing the analyses with quantitative evidence made that clear, and I have removed those claims, and significantly changed the discussion so as not to make correct interpretations based on the data.

Minor Points

P3 L31: It is debatable whether P is the *ultimate* limiting nutrient in freshwater systems. This supposition is not supported by your reference to Sterner 2008 and certainly not by the meta analyses of Elser et al. 2007 or any of the multitude of studies examining N & P co-limitation conducted in the last decade or so. It certainly is an important limiting element, but not necessarily the *most* important.

I have edited this line to reflect that P is one of the important limiting nutrients.

P3 L30: It's true that daphnids could be more susceptible to P-limitation, but as most environments are experiencing P-loading rather than P reductions it is unclear to me how they might be used as an indicator organism. Wouldn't it make more sense to use an organism sensitive to excess P?

Yes, I have removed the information about daphnia being an indicator organism as they indicate for other stressors, not P-limitation.

P4 L12: "Clonal variation can be considered as important as species identity" Please explain what this importance is referring to.

I have clarified this part of the introduction to give more information about why clonal variation may contribute a significant amount to variation as well (or potentially instead of) species variation.

P5 L5: Are these predictions here? Do they all refer to the final study question 3? If not, you might consider moving them up with their corresponding question or spelling out some of these connections specifically in respect to question 3 earlier in the paragraph bc this information could be confusing out of context.

Those were predictions. I have moved this information so that they would not be out of context.

P6 L3: Another caveat that you might want to add is that these animals have been adapted to lab conditions for long periods of time which might influence the magnitude of their plastic responses compared to animals in the wild.

Added as suggested.

P6 L47: So the total amount fed to these animals would be 0.5 mg C day? Can you show that this level of food is not limiting to the larger *magna* and *pulex* species given that there were also likely neonates feeding in these tubes later on in the experiment? Could this account for some of the trait variance in these larger body sized animals?

The total amount fed would be 1.0 mg C, as they were individually housed in jars. I have included a citation to show that this amount of food should not be limiting in quantity (Sterner and Robinson 1994).

P7 L38-45: These sentences might be better off in the results section.

Edited as suggested (see supplemental material)

P9 L12: Do you have quantitative statistics to support this? I only see descriptive statistics for individual traits in the tables and figures.

I have added more statistical tests in order to provide more backing for conclusions. See general comments to the editor.

P9 L38: This shift might be moved to the discussion where it can be discussed further.

This information was moved to the supplementary material.

P12 L34: It's not clear how these phylogenetic differences could've been related to plasticity. The work by Seidendorf 2010 doesn't support this, but if you believe that this might be the case, would it be worth including it as a covariate in your analyses?

Thank you for this comment so that I can correct this mistake. I have removed any claims to potential phylogenetic influences on plasticity. The quantitative analysis that I have done does not indicate that there is support for this.

P12 L51: You might consider changing this to "partially matches the prediction". There might be an individual *magna* and *pulex* clone that is more sensitive, but on average *obtusa* seems to be the most sensitive species.

This has also been removed due to subsequent analyses.

P13 L19-21: It would be really helpful to develop this idea in the introduction.

Added as suggested. There is more information in the introduction about a tradeoff between trait variation and responses to environmental change.

P13 L38: Hood and Sterner 2014 didn't measure clutch frequency or size.

Thank you for the correction. I had referenced the paper in their use of flexibility in resource use. The claim for differences in clutch size and frequency were from my own data. That has been edited (actually removed altogether).

P13 L 43: I really like this idea, so if you get a chance you could expand upon it in future versions of this manuscript or if not in future work.

Thank you. I am not able to draw any further conclusions based on what I have done, but I will keep it in mind for future work.

Tables and Figures: It might help if you rearrange these to put large- and small-bodied animals together.

The legends for Figure 3 could also do with a bit of reformatting.

Figure 3 has been moved to the supplemental material. As less of my results/conclusions/interpretations rely much less on daphniid body size, I have not placed small-bodied, and large-bodied species together.

Appendix C

VARIATION IN LIFE-HISTORY TRAITS AMONG *DAPHNIA* AND ITS RELATIONSHIP TO
SPECIES-LEVEL RESPONSES TO PHOSPHORUS LIMITATION

Rachel Hartnett¹

¹Department of Biology, Program in Ecology and Evolutionary Biology

The University of Oklahoma

730 Van Vleet Oval, Room 314

Norman, O.K. 73019

U.S.A.

Abbreviated title: Variation and Responses During Food Stress

Correspondence: Rachel Hartnett, Department of Biology, University of Oklahoma, 730 Van Vleet Oval, E Norman,
OK, USA 73019. Email: rhartnett@ou.edu

Abstract

Currently organisms are experiencing changes in their environment at an unprecedented rate. Therefore the study of the contributions to and responses in traits linked to fitness is crucial, as they have direct consequences on a population's success in persisting under such change. *Daphnia* is used as a model organism as the genus contains keystone primary consumers in aquatic food webs. A life-history table experiment (LHTE) using four species of *Daphnia* was conducted to compare variation in life-history traits among species across two different environmental conditions (high and low phosphorous availability). Results indicate that the food quality environment had the most impact on life-history traits, while genetic contributions to traits were higher at the species-level than clonal-level. Higher trait variation and species-level responses to P-limitation was more evident in reproductive traits, while growth traits were found to be less affected by food quality and had less variation. Exploring trait variation and potential plasticity in organisms is increasingly important to consider as a potential mechanism for population persistence given the fluctuations in environmental stressors we are currently experiencing.

Keywords: zooplankton, stoichiometry, body-size, intraspecific variation, phosphorous

Background

With increasing environmental stress, many suites of organismal traits are expected to experience strong selection, with life-history traits potentially being among the most impacted (Bradshaw and Holzapfel 2008, Reed *et al.* 2011). Life-history traits have a direct link to fitness, as an organism's success is built upon an ability to grow to reproductive age, the timing of reproduction events, as well as cumulative reproductive output. Therefore, life-history theory has established direct associations between a population's environment and life-history trait evolution (Stearns 1992, Agrawal *et al.* 2013).

Food stress has been shown to create a variety of life-history trait effects in organisms, which include longer developmental time, decreases in body size, and lowered fecundity (Ellers & Van Alphen 1997; Nylin & Gotthard 1998). Food stress can be experimentally manipulated through decreasing a limiting resource. In many freshwater lentic systems, phosphorus (P) is one of the limiting nutrients (Wetzel 1983, Sterner 2008), with anthropogenic inputs of P in aquatic systems forcing rapid change in zooplankton populations (Frisch *et al.* 2014). P-limitation (i.e., low food quality) has effects on *Daphnia* life-history traits such as growth, reproduction, and senescence (e.g., Dudycha 2003, Jeyasingh and Weider 2005). Members of the genus *Daphnia* (Cladocera: Anomopoda) have one of the highest P contents amongst zooplankton, so they are predicted to be more responsive to P-limitation compared to other zooplankton taxa (Sterner and Schulz 1998); however P content alone is not sufficient to predict species-level responses in growth (DeMott and Pape 2004).

Daphnia have a cyclically parthenogenic life-cycle, which includes bouts of asexual reproduction under good growing conditions, and sexual reproduction during times of food stress, changes in photoperiod, and crowding cues (Kleiven *et al.* 1992). This creates the unique advantage of establishing multiple clonal lineages in a population leading to the maintenance of high genetic variation in many natural *Daphnia* populations (Innes *et al.* 1986; Spitze *et al.* 1991;

Weider *et al.* 1999). In addition, researchers have found ~~large~~ strong clonal responses to predator cues (Spitze 1992, Weider and Pijanowska 1993), nutrient limitation (Lynch 1989, Weider *et al.* 2004), habitat selection (de Meester 1994) and toxins (Baird *et al.* 1990, Walls 1997). Intraspecific genetic variation has been shown to have effects on population-level processes like colonization (Crutsinger *et al.* 2008, Crawford and Whitney 2010), coexistence (Lankau *et al.* 2009), and predation (Post *et al.* 2008). For *Daphnia*, clonal diversity is better maintained under P-limitation (Weider *et al.* 2008), therefore clonal variation may significantly contribute to overall add to genetic contributions variation among *Daphnia* species.

Formatted: Font: Italic

Under the hypothesis of environmental buffering, influential life-history traits should have minimal trait variation, as fitness would be heavily dependent on minimal change within important vital rate constraints (Pfister 1998). Over multiple life-history tables, Pfister (1998) showed that traits with the highest variation had the least response to temporal change. This could lead to a demonstrates a potential tradeoff between trait variation and species-level responses to environmental change, where traits that are more vital to fitness would show less trait variation. However, ~~life-history traits could~~ vary in the amount of variation seen under food stress; P-limitation has been shown to differentially affect somatic growth rates and reproductive investment like in egg size and abortion rates within *Daphnia* due to differing nutritional requirements (Hood and Sterner 2014). ~~This could lead to a potential tradeoff between trait variation and species-level responses to environmental change. Increased intraspecific variation could allow for flexibility at the species level, and potentially mediate environmental effects on life history traits. Alternatively, trait variation and species-level responses to environmental changes could be positively correlated, as traits with low variation may have a lower range of trait values that are measurable in species-level responses (i.e., a floor effect), and it has been documented that more intraspecific trait variation also leads to higher variation in intraspecific responses across environments (e.g., Siefert~~

and Ritchie 2016). This trend between intraspecific trait variation and intraspecific responses could translate to species-level responses as well.

Another mechanism to mitigate environmental effects is an organism's body size. *Daphnia*'s physiology allow them to alter filtering rates under different food quality environments (Sahuquillo *et al.* 2007), although there are phylogenetic constraints. Jeyasingh (2007) suggested that evolution should favor more responsive physiologies for smaller organisms in order to counter frequent shifts in nutrient limitation. Smaller aquatic organisms use phosphorus at higher rates (Johannes 1964), which means that they would experience P-limitation more frequently. This could lead to smaller-bodied species becoming more plastic than larger-bodied species.

This present study aims to address the following: 1) how much does the environment vs. genetic (taxonomic) identity contribute to life-history trait variation? Do clonal lines contribute more to trait variation than species identity? 2) Do smaller species exhibit more plastic physiologies? And 3) what is the potential relationship between intraspecific variation and species-level responses to food stress?

Methods

Study organism

Daphnia is a cosmopolitan genus (Sarma *et al.* 2005, Lampert 2011). Three clonal lineages from each of four different *Daphnia* species (*D. magna*, *D. mendotae*, *D. obtusa*, and *D. pulex*) were collected from a variety of laboratory stocks (see Table 1). These clonal lineages span the three subgenera of *Daphnia*, ranging across North America and Europe, and come from various aquatic habitats (Table 1). *D. magna* clones used in this study originated from South Dakota, Finland and Germany from a spectrum of habitats. The South Dakotan clone (MA3) came from a permanent lake, a shallow (< 2 m) prairie pot-hole (Weider *et al.* 2004). MA2 and MA1 are both inbred lines from an original genetic cross between a Finnish clone and a German clone. MA2 was inbred for

three generations and MA1 was inbred for one generation (Dieter Ebert, Switzerland, personal communication). The environment of the parental clones include a Finnish clone from a ephemeral pond with desiccation in spring/summer and freezing during autumn/winter and a clone from a German semi-permanent pond, with freezing in the winter (Roulin *et al.* 2013). In addition, *D. pulex* and *D. obtusa* clones came from temporary ponds in the U.S. Midwest, while *D. mendotae* came from permanent lakes in the U.S. Midwest. One *D. mendotae* clone (ME3) experienced high levels of mortality early on in the experiment, and was subsequently dropped from the analyses. These contrasting environments have created very different evolutionary trajectories for these species. However, a couple of caveats should be noted: these lineages have been adapted to lab conditions and may have different capacities in variation and response than animals collected directly from the field, and a potential confounding issue due to inbreeding could affect the variation and response for two of the three *D. magna* clones (MA1 and MA2).

Experimental design

Clonal lineages were maintained as separate populations in 900 mL jars, with regular and plentiful feeding using the chemostatically-cultured green algae, *Scenedesmus acutus*, at a constant 20°C in COMBO media (Kilham *et al.* 1998). These stock cultures maintained an approximate density of 12-30 adults L⁻¹. A small amount of cetyl alcohol (~10 mg) was added to act as a surfactant to prevent animals from being trapped in the air-water interface. Stock cultures and experimental animals received equal amounts of 24-hour incidental ambient lighting. Maternal lines for experimental animals were raised individually in 60 mL jars with 50 mL of COMBO and fed daily with 1 mg C L⁻¹ of *S. acutus* that was grown in nutrient-rich conditions (i.e., C:P, ~100:1). The volume amount of *S. acutus* added each day for 1 mg C L⁻¹ was calculated by measuring the absorbance of chlorophyll at wavelengths 660nm and 740nm using a spectrophotometer (Spectronic@20 Genesis, Madison, WI) and using a calibrated chlorophyll-carbon correlation to

~~convert the value to amount of carbon~~ curve (Sterner 1993). Females were monitored every 24-hours, and first and second clutches were removed. Experimental animals (N = 20 per clone) were taken from third or later clutches of these individually raised maternal lines within 24 hours to reduce maternal effects (Ebert 1991). Experimental animals needed to be pooled from multiple mothers in order to reach the appropriate sample size.

Starting on August 17th, 2013, An initial body-length measurement (i.e., start length) of each experimental animal was taken using a MOTICAM 2300 digital camera and software system (Motic®, S-05165) mounted to an Olympus BX51 compound dissecting microscope. Length measurements were taken from the top of the eyespot to the base of the core body, right above the top of the tail-spine. The tail-spine is known to be morphologically plastic depending on environmental conditions, and was not measured with core length due to potential confounding length measurements. Then, experimental animals were placed individually into 60 mL glass jars with 50 mL of COMBO at 20°C, and were divided into two environmental conditions, high and low phosphorus (N=10 per clonal line for each environmental treatment). Nutrient media ~~was~~ prepared using a modification of COMBO (Kilham *et al.* 1998), with additions of sodium nitrate and potassium phosphate. Animals under a high phosphorus (HiP) feeding regime were fed daily with 1 mg C L⁻¹ of *S. acutus* that was grown in nutrient-rich conditions (i.e., 85.01 mg L⁻¹ sodium nitrate and 8.71 mg L⁻¹ potassium phosphate and C:P ~100:1). A low phosphorus (LoP) feeding regime consisted of daily 1mg C L⁻¹ feeding of *S. acutus* grown in nutrient-poor conditions (i.e., 42.5 mg L⁻¹ sodium nitrate and 0.87 mg L⁻¹ potassium phosphate and C:P ~750:1). These food conditions should not be limiting in quantity (Sterner and Robinson 1994). Experimental animals were transferred every two days to fresh jars in order to avoid carbon (detrital) accumulation that could differentially affect resource availability based on inter-/intra-specific variation of filtering rates. Experimental animals were monitored daily and size was measured again at maturation, when first egg development was seen (i.e., age at maturation and length at maturation). Clutch size

Formatted: Superscript

was recorded daily, as well as images for neonate body-lengths ($N \leq 5$ neonates per clutch in order to reduce small-clutch bias). Number of clutches, clutch size and mean neonate length (termed mean clutch length) were calculated from these daily recorded measurements. Dead experimental animals were measured with the day of death. The experiment ran for 28 days, and at the end of this period, experimental animals were measured (i.e., end length), as described above.

Statistical Analyses

All analyses were ~~run~~ conducted using R (R core team, 2018) unless specified otherwise in the methods. Required packages for this analysis included: dplyr (Wickham *et al.* 2019), ggplot2 (Wickham 2016), ggpubr (Kassambara 2018), and Hmisc (Harrell *et al.* 2019). Individuals (replicates) were dropped from analysis if they died within 5 days of the start of the experiment to prevent bias from missing data in reproductive measurements. Data was screened for outliers and cases were removed if they were ± 2 standard deviations from a mean value. Mortality accounts for less than five percent of the study, and a chi-square contingency analysis was ~~run~~ conducted between species and treatment to ensure that survivorship was independent of species identity or treatment. A MANOVA was conducted to measure the effects of genetic (species, clonal) and environmental (phosphorous treatment) contributions on life-history traits (length at maturation, age at maturation, end length, mean clutch length, mean clutch size, and number of clutches). The start length of individuals was used as a covariate, as well as maternal line; maternal effects are common among daphniid studies (Lampert 1993), so maternal line was also used as a potential confounding variable. Multivariate normality was checked for the dataset using post-hoc residuals from the MANOVA, and the proportion of variance explained for each main effect was calculated from Wilks' lambda, where $partial \eta^2 = 1 - \lambda^{1/s}$; λ is Wilks' lambda for the main effect and $s = \min(6, df_{effect})$ (Tabachnick & Fidell 2013). The MANOVA was ~~run~~ conducted using SPSS (Version 20, IBM).

In order to identify differences in species-level responses to food stress, multiple linear regressions were ~~run~~ conducted for each trait, with food treatment and species as factors. The significance-level was adjusted using the Bonferroni correction (alpha = 0.008). Intraspecific variation was measured for each species by coefficients of variation (COVs). COVs were calculated from $CV = \frac{s}{\bar{y}}$, where s = standard deviation and \bar{y} = mean of the particular life-history trait for that species. Thus, COVs measure the amount of variation within the species for each trait. Species-level responses to food stress were quantified by using the differences in log-transformed values between phosphorous treatments (Seidendorf *et al.* 2010): Species-level response per trait = $\ln(\text{trait}_{\text{HIP}}) - \ln(\text{trait}_{\text{LoP}})$ for each species. Individual t-tests were ~~run~~ conducted on species-level responses and COVs, respectively, between the larger species (*D. magna* and *D. pulex*) compared to the smaller species (*D. mendotae* and *D. obtusa*). A Pearson correlation between COVs and these species-level response values were also ~~run~~ carried out to test the relationship between trait variation and the magnitude of response to food stress. Separate correlations were ~~run~~ conducted per species and per trait; the significance-level was adjusted using the Bonferroni correction (alpha = 0.002).

Results

Mortality in this study was independent of species identity and food treatment (chi-square contingency, $df = 3$, $\chi^2 = 0.1361$, $p = 0.9872$). Descriptively, under low-phosphorous (LoP) conditions, all clonal lines of all species showed smaller sizes both at first reproduction, and at the end of the experiment. Similarly, under LoP, clones exhibited delayed onset of reproduction and had smaller clutch sizes. The number of clutches varied per clone, as well as their mean clutch length (See S1). Visual inspection of trait variation mapping using a PCA indicated that species showed more change in reproductive traits rather than size traits between food treatments (See S2).

The MANOVA showed that both genetic factors (species, clone) as well as environmental factors (maternal effects, food treatment) significantly affected life-history traits (Table 2). The length of individuals at the start, the blocking effect of time, and maternal effects did not significantly affect life-history traits (Table 2). The proportion of variance explained can be used as a proxy for the magnitude of effect size metric (Tabachnick & Fidell 2013). Using the *partial* η^2 statistic as a relative effect size metric, food quality environment explained the greatest proportion of variance (*partial* $\eta^2 = 0.85$, $df_{\text{effect}} = 1$), while genetic factors explained less; species identity (*partial* $\eta^2 = 0.60$, $df_{\text{effect}} = 3$) explained more than the clonal level (*partial* $\eta^2 = 0.23$, $df_{\text{effect}} = 7$) (Table 2). Therefore, the environment had a stronger main effect than either genetic component, with species identity having a stronger effect than clonal identity.

The size of the species did not show significant differences in species-level responses (two-samples t-test, $t = 0.75$, $df = 22$, $p > 0.1$), ~~but~~ However, they did show significant differences in the amount of intraspecific variation among traits, with larger species showing more variation than smaller species (two-samples t-test, $t = 8.41$, $df = 22$, $p < 0.01$).

All life-history traits were impacted by both species identity and the food quality level (length at maturation, adjusted $R^2 = 0.921$, age at maturation adjusted $R^2 = 0.670$, end length, adjusted $R^2 = 0.752$, mean clutch length, adjusted $R^2 = 0.146$, mean clutch size, adjusted $R^2 = 0.708$, number of clutches, adjusted $R^2 = 0.464$; all adjusted $p < 0.01$). Effects of food quality were detected across all species for some reproductive traits (length at maturation and clutch size), while mean clutch length only had effects for larger species *D. magna* and *D. pulex* (Figure 1). *D. mendotae* was the only species that had an effect with food quality treatment on the length of individuals at the end of the experiment (Figure 1).

Intraspecific trait variation and species-level responses were significantly positively correlated ($r^2 = 0.89$, $t = 9.60$, $df = 22$, $p < 0.01$) (Figure 2). Correlations ~~conducted~~ per species

found that *D. magna* and *D. obtusa* had significant correlations between intraspecific trait variation and species-level responses (*D. magna*: $r^2 = 0.94$, $t = 5.51$, $df = 4$, $p < 0.001$; *D. obtusa*: $r^2 = 0.99$, $t = 12.151$, $df = 4$, $p < 0.001$). There was no significant correlations detected when ~~running analyzing~~ correlations per trait.

Discussion

The environment contributed most to life-history traits and species identity ~~was the~~ ~~larger explained greater trait variation than genotype genetic contributor~~

A keystone of evolutionary theory is that trait variation can be separated into genetic and environmental components; when a selection pressure ~~is imposed~~~~imposes~~ on a trait, the environmental variance will dominate over genetic variance due to reduction in genetic variance in the selection process (Spitze 1991). The food quality available in the environment had the largest effect on life-history traits, accounting for 85% of the variance (Table 2). Food quality affected all species current reproduction via clutch size, while it only had significant effects on mean clutch length, which could have implications for fitness of future generations, for larger-bodied *D. magna* and *D. pulex*. Food quality is known to play an important role in growth rate, many studies have used the growth rate hypothesis to predict relationships between growth and P-limited diets or environments (e.g., Jeyasingh 2007, Hood and Sterner 2014). P-limitation in particular has been well studied in *Daphnia*; P-intensive processes including organismal growth, rRNA levels, and therefore protein production, require high availability of phosphorus (Weider *et al.* 2004).

Depending on intra- and inter-specific pressures, evolution will favor more or less specialized individuals within a generalist population (Araújo *et al.* 2011). Within these populations, clonal variation could lead to a wider environmental range in which the population can maintain fitness under environmental change (Nunney 2015). Clonal lineages did not have a

large contribution to life-history traits, as predicted. This potentially could be due to the relatively small number of clones per species. In addition, these experimental clonal assemblages are somewhat of an artificial construct and raised in the lab, which may not reflect field populations. It is enticing that species identity clustered strongly in terms of composite life-history traits (i.e., PC axes) under high quality (high P) food conditions, but there was no strong species-specific clustering among three of the species under poor nutrient conditions (S2).

Trait variation in body size is constrained while there is flexibility in reproductive traits

The food quality levels from this study did not support the notion that smaller-bodied *Daphnia* respond more to changes in food quality than larger-bodied *Daphnia*. However, larger *Daphnia* did show more variation in life-history traits. Body size traits were associated with lower amounts of intraspecific variation and species-level responses to food quality, while other reproductive traits were associated with higher amounts of variation and responses to food quality (Figure 2). This indicates that *Daphnia* have size-based phenotypes that are somewhat genetically constrained. Allometric constraints may be one possible explanation for conserved morphological traits. It has been shown that regardless of body-size, daphniids all follow a similar pattern of resource allocation to growth and reproduction under different levels of food (carbon) quantity (Dudycha & Lynch 2005).

Body size has been implicated in determining sensitivity to food quality, with larger individuals being affected by low food quality more so than smaller individuals (Peter and Lampert 1989). However there was no evidence in this study that the body-size of the species played a role in species-level responses or the amount of intraspecific variation. Previous work has indicated that *Daphnia* may face a competitive tradeoff between maximizing growth when food quality is high and minimizing negative effects of poor food quality (Seidendorf *et al.* 2010, Hood and Sterner 2014). However, stoichiometric flexibility may allow for changing the C:P ratio of

new growth under P-limited conditions in order to avoid consequences of P-limited diets. *D. mendotae* is relatively inflexible compared to the other species in this study in its response to changes in the C:P of its diet, showing strong homeostasis between diets (Hood and Sterner 2014). This matches with this study's results, where *D. mendotae* had overall less intraspecific variation across traits, and smaller species-level responses (Figure 2).

We hypothesized a tradeoff between intraspecific trait variation and species-level responses to food stress. However, in this present study, results were contradictory to expectations: species showed more change between food quality environments with increasing trait variation. This is most likely due to reproductive traits being very P-intensive and very responsive to changes in food quality. In *Daphnia*, P and reproductive trait relationships have not been as well studied as somatic growth rate (SGR), a well-known proxy of fitness (Lampert & Trubetskova 1996). However, shifts toward lower reproduction has been seen for low levels of nitrogen and phosphorus (Sterner *et al.* 1992) and for low food concentrations (Lynch 1989); but under toxin-enriched environments, *Daphnia* have shown to maintain reproductive output (Forbes *et al.* 2016). Plasticity in reproductive traits are generally considered less important in changing population growth rates based on previous modeling of growth and reproductive schedules (Pfister 1998). These results suggest that potentially environmental buffering from P-limitation has canalized the highly vital growth traits over time, while leaving plasticity in reproductive rates variable and responsive to environmental change. However, another potential explanation could be that a tradeoff would not be expected in this case because these lineages have been raised in the lab for many generations with bountiful resources, thereby creating a so-called "superflea" (Tessier #). Reznick et al. 2000. This study should be repeated using individuals taken from natural habitats.

Conclusions

This present study provides evidence that species identity is important in determining life-history traits, but that may not translate into size-structured populations due to variation in reproductive traits across

environments that vary in overall food quality. In particular, the flexibility in reproductive traits may play an important role for population persistence in the face of environmental change. Phenotypic plasticity is the ability of an organism to change its phenotype in response to environmental change. *Daphnia* have shown a great capacity for phenotypic plasticity in predator-avoidance (e.g., Spitze 1992; Weider & Pijanowska 1993), nutrient uptake/use efficiency (e.g., Lampert 1994), and other life-history traits (e.g., Lampert 1993). This study, where a changing environment may select for more responsiveness in reproductive traits, indicates more consideration of evolution of phenotypic plasticity and population persistence through life-history traits (Chevin et al. 2010). Gathering information about the potential for phenotypically plastic traits via trait variation has been, and will continue to be, a goal toward predicting a species' ability to respond to continued environmental stress. However, there are costs and limits involved in maintaining plastic traits, including genetic and/or developmental constraints, competitive exclusion by a more optimal (and less plastic) trait during a stable period, or geographical limits (Whitlock 1996, Pigliucci 2005). Species that are flexible in their use of phosphorus may compensate for P-limitation by being more plastic in reproductive life-history traits.

Ethics statement

No special research ethical approval was required at that time. No special fieldwork or collecting permissions were required at that time. No special animal ethical care permissions were required at that time. All applicable institutional and/or national guidelines for the care and use of animals were followed.

Data Accessibility

Data is deposited at Dryad and available for review at <https://datadryad.org/review?doi=doi:10.5061/dryad.d4v7g74>.

Competing Interests

The author declares that she has no competing interests.

Author Contributions

RH is the sole author and contributed the concept, experimental design, sample and data collection, data analysis, and manuscript preparation.

Funding

This work was supported by a US National Science Foundation Graduate Research Fellowship Program under Grant No. 2013151892.

Acknowledgements

I thank my graduate advisor, L.J. Weider, for doing his job extremely well. Special thanks to M. Pfrender, L. Orsini, and K. Spitze for providing clonal lineages for *D. magna*, *D. pulex*, and *D. obtusa*. I deeply appreciate the input from L.J. Weider, K. Spitze, J. Dudycha, four anonymous reviewers, and J. Beyer, whose comments improved this manuscript immensely. Also, I would like to thank K. Roeder, B. Tweedy, E. Freitas, and E. Kiehnau for reviewing earlier drafts of the manuscript. This manuscript constitutes a portion of R. Hartnett's dissertation in partial fulfillment of the Ph.D. requirements at the U. of Oklahoma.

Funding

~~This work was supported by a US National Science Foundation Graduate Research Fellowship Program under Grant No. 2013151892.~~

References

- Agrawal, A.A., Johnson, M.T., Hastings, A.P., Maron, J.L. & Reznick, S.E. (2013). A field experiment demonstrating plant life-history evolution and its eco-evolutionary feedback to seed predator populations. *Am. Nat.*, 181, 35–45.
- Araújo, M.S., Bolnick, D.I. & Layman, C.A. (2011). The ecological causes of individual specialisation. *Ecol. Lett.*, 14, 948–958.
- Baird, D.J., Barber, I. & Calow, P. (1990). Clonal variation in general responses of *Daphnia magna* Straus to toxic stress. I. Chronic life-history effects. *Funct. Ecol.*, 4, 399–407.
- Bradshaw, W.E. & Holzapfel, C.M. (2008). Genetic response to rapid climate change: it's seasonal timing that matters. *Mol. Ecol.*, 17, 157–166.
- Chevin, L. M., Lande, R. & Mace, G.M. (2010). Adaptation, plasticity, and extinction in a changing environment: towards a predictive theory. *PLOS Biol.*, 8, e1000357.
- Crawford, K.M. & Whitney, K.D. (2010). Population genetic diversity influences colonization success. *Mol. Ecol.*, 19, 1253–1263.

- Crutsinger, G.M., Souza, L. & Sanders, N.J. (2008). Intraspecific diversity and dominant genotypes resist plant invasions. *Ecol. Lett.*, 11, 16–23.
- DeMott, W.R., & Pape, B. (2004). Stoichiometry in an ecological context: testing for links between *Daphnia* P-content, growth rate and habitat preference. *Oecologia*, 142, 20-27.
- Dudycha, J.L. (2003). A multi-environment comparison of senescence between sister species of *Daphnia*. *Oecologia*, 135, 555–563.
- Dudycha, J.L. & Lynch, M. (2005). Conserved ontogeny and allometric scaling of resource acquisition and allocation in the Daphniidae. *Evolution*, 59, 565–576.
- Ebert, D. (1991). The effect of size at birth, maturation threshold, and genetic differences on the life-history of *Daphnia magna*. *Oecologia*, 86, 243–250.
- Ellers, J. & Van Alphen, J.J.M. (1997). Life history evolution in *Asobam tabida*: plasticity in allocation of fat reserves to survival and reproduction. *J. Evolution Biol.*, 10, 771–785.
- Forbes, V.E., Galic, N., Schmolke, A., Vavra, J., Pastorok, R. & Thorbek, P. (2016). Assessing the risks of pesticides to threatened and endangered species using population modeling: A

critical review and recommendations for future work. *Environ. Toxicol. Chem.*, 35, 1904–1913.

Frisch, D., Morton, P.K., Chowdhury, P.R., Culver, B.W., Colbourne, J.K., Weider, L.J., & Jeyasingh, P.D. (2014). A millennial-scale chronicle of evolutionary responses to cultural eutrophication in *Daphnia*. *Ecol. Lett.*, 17, 360–368.

Harrell Jr., F.E., with contributions from Charles Dupont and many others. (2019). Hmisc: Harrell Miscellaneous. R package version 4.2-0. <https://CRAN.R-project.org/package=Hmisc>

Hartnett RN. Data from Variation in life-history traits among *Daphnia* species and its relationship to species-level responses to phosphorus limitation. Dryad Digital Repository. <https://doi.org/10.5061/dryad.d4v7g74>.

Hood, J.M. & Sterner, R.W. (2014). Carbon and phosphorus linkages in *Daphnia* growth are determined by growth rate, not species or diet. *Funct. Ecol.*, 28, 1156–1165.

Innes, D.J., Schwartz, S.S. & Hebert, P.D. (1986). Genotypic diversity and variation in mode of reproduction among populations in the *Daphnia pulex* group. *Heredity*, 57, 345–355.

Jeyasingh, P.D. & Weider, L.J. (2005). Phosphorus availability mediates plasticity in life-history traits and predator–prey interactions in *Daphnia*. *Ecol. Lett.*, 8, 1021–1028.

Jeyasingh, P.D. (2007). Plasticity in metabolic allometry: the role of dietary stoichiometry. *Ecol. Lett.*, 10, 282–289.

Kassambara, A. (2018). ggpubr: 'ggplot2' Based Publication Ready Plots. R package version 0.2.
<https://CRAN.R-project.org/package=ggpubr>

Kilham, S.S., Kreeger, D.A., Lynn, S.G., Goulden, C.E. & Herrera, L. (1998). COMBO: a defined freshwater culture medium for algae and zooplankton. *Hydrobiologia*, 377, 147–159.

Kleiven, O.T., Larsson, P. & Hobæk, A. (1992). Sexual reproduction in *Daphnia magna* requires three stimuli. *Oikos*, 65, 197–206.

Lampert, W. (1993). Phenotypic plasticity of the size at first reproduction in *Daphnia*: the importance of maternal size. *Ecology*, 74, 1455–1466.

Lampert, W. (1994). Phenotypic plasticity of the filter screens in *Daphnia*: adaptation to a low-food environment. *Limnol. Oceanogr.*, 39, 997–1006.

Lampert, W. & Trubetskova, I. (1996). Juvenile growth rate as a measure of fitness in *Daphnia*. *Functional Ecology*, 10, 631–635.

Lampert, W. (2011). *Daphnia: development of a model organism in ecology and evolution*. Excellence in Ecology; 21. International Ecology Institute, Oldendorf, Germany.

Lankau, R.A., Ackerly, A.E. & DeAngelis, E.D.L. (2009). Genetic variation promotes long-term coexistence of *Brassica nigra* and its competitors. *Am. Nat.*, 174, E40–E53.

Lynch, M. (1989). The life history consequences of resource depression in *Daphnia pulex*. *Ecology*, 70, 246–256.

Meester, L.D. (1994). Life histories and habitat selection in *Daphnia*: divergent life histories of *D. magna* clones differing in phototactic behaviour. *Oecologia*, 97, 333–341.

Nunney, L. (2015). Adapting to a changing environment: modeling the interaction of directional selection and plasticity. *J Heredity*, 107, 15–24.

Nylin, S. & Gotthard, K. (1998). Plasticity in life-history traits. *Annu. Rev. Entomol.*, 43, 63–83.

Peter, H. & Lampert, W. (1989). The effect of *Daphnia* body size on filtering rate inhibition in the presence of a filamentous cyanobacterium. *Limnol. Oceanogr.*, 34, 1084–1089.

Pfister, C.A. (1998). Patterns of variance in stage-structured populations: Evolutionary predictions and ecological implications. *P Natl. Acad. Sci-Biol.*, 95, 213–218.

Pigliucci, M. (2005). Evolution of phenotypic plasticity: where are we going now? *Trends in Ecol. Evol.*, 20, 481–486.

Post, D.M., Palkovacs, E.P., Schielke, E.G. & Dodson, S.I. (2008). Intraspecific variation in a predator affects community structure and cascading trophic interactions. *Ecology*, 89, 2019–2032.

R Core Team (2018). R: A language and environment for statistical computing. R Foundation for Statistical Computing, Vienna, Austria. URL <https://www.R-project.org/>.

Reed, T.E., Schindler, D.W. & Waples, R.S. (2011). Interacting effects of phenotypic plasticity and evolution on population persistence in a changing climate. *Conserv. Biol.*, 25, 56–63.

Reznick, D., Nunney, L. & Tessier, A. (2000). Big houses, big cars, superfleas and the costs of reproduction. *Trends Ecol. Evol.*, 10, 421-425.

Formatted: Font: Italic

Roulin, A.C., Routtu, J., Hall, M.D., Janicke, T., Colson, I., Haag, C.R. & Ebert, D. (2013).

Local adaptation of sex induction in a facultative sexual crustacean: insights from QTL mapping and natural populations of *Daphnia magna*. *Mol. Ecol.*, 22, 3567–3579.

Sahuquillo, M., Melão, M.G.G. & Miracle, M.R. (2007). Low filtering rates of *Daphnia magna*

in a hypertrophic lake: laboratory and *in situ* experiments using synthetic microspheres.

Hydrobiologia, 594, 141–152.

Sarma, S.S.S., Nandini, S. & Gulati, R.D. (2005). Life history strategies of cladocerans:

comparisons of tropical and temperate taxa. *Hydrobiologia*, 542, 315–333.

Seidendorf, B., Meier, N., Petrusek, A. Boersma, M., Streit, B. & Schwenk, K. (2010).

Sensitivity of *Daphnia* species to phosphorus-deficient diets. *Oecologia*, 162, 349-357.

Siefert, A. & Ritchie, M.E. (2016). Intraspecific trait variation drives functional responses of old-

field plant communities to nutrient enrichment. *Oecologia*, 181, 245-255.

Formatted: Font: Italic

Spitze, K. (1991). *Chaoborus* predation and life-history evolution in *Daphnia pulex*: temporal

pattern of population diversity, fitness, and mean life history. *Evolution*, 45, 82-89.

Spitze, K., Burnson, J. & Lynch, M. (1991). The covariance structure of life-history characters in *Daphnia pulex*. *Evolution*, 45, 1081–1090.

Spitze, K. (1992). Predator-mediated plasticity of prey life history and morphology: *Chaoborus americanus* predation on *Daphnia pulex*. *Am. Nat.*, 139, 229–247.

Stearns, S.C. (1992). *The evolution of life histories*. Oxford University Press, Oxford ; New York.

Sterner, R.W., Elser, J.J. & Hessen, D.O. (1992). Stoichiometric relationships among producers, consumers and nutrient cycling in pelagic ecosystems. *Biogeochemistry*, 17, 49–67.

Sterner, R.W. (1993). *Daphnia* growth on varying quality of *Scenedesmus*: mineral limitation of zooplankton. *Ecology*, 74, 2351-2360.

Sterner, R. W., & Robinson, J.L., (1994). Thresholds for growth in *Daphnia magna* with high and low phosphorus diets. *Limnology and Oceanography*, 39, 1228-1232.

Sterner, R.W. & Schulz, K.L. (1998). Zooplankton nutrition: recent progress and a reality check. *Aquat. Ecol.*, 32, 261–279.

Formatted: Font: +Body (Calibri)

Formatted: Normal, Indent: Left: 0 pi, First line: 0 pi

Formatted: Font: Italic

Sterner, R.W. (2008). On the phosphorus limitation paradigm for lakes. *International Review of Hydrobiology*, 93, 433–445.

Tabachnick, B.G. & Fidell, L. (2013). *Using multivariate statistics*. 6th ed. Pearson Education, Boston.

Walls, M. (1997). Phenotypic plasticity of *Daphnia* life history traits: the roles of predation, food level and toxic cyanobacteria. *Freshw. Biol.*, 38, 353–364.

Weider, L.J. & Pijanowska, J. (1993). Plasticity of *Daphnia* life histories in response to chemical cues from predators. *Oikos*, 67, 385–392.

Weider, L.J., Hørbæk, A., Hebert, P.D.N. & Crease, T.J. (1999). Holarctic phylogeography of an asexual species complex – II. Allozymic variation and clonal structure in Arctic *Daphnia*. *Mol. Ecol.*, 8, 1–13.

Weider, L.J., Glenn, K.L., Kyle, M. & Elser, J.J. (2004). Associations among ribosomal (r)DNA intergenic spacer length, growth rate, and C:N:P stoichiometry in the genus *Daphnia*. *Limnol. Oceanogr.*, 49, 1417–1423.

Weider, L.J., Jeyasingh, P.D. & Looper, K.G. (2008). Stoichiometric differences in food quality: impacts on genetic diversity and the coexistence of aquatic herbivores in a *Daphnia* hybrid complex. *Oecologia*, 158, 47–55.

Wetzel, R.G. (1983). *Limnology*. 2nd ed. Saunders, Philadelphia.

Whitlock, M.C. (1996). The red queen beats the jack-of-all-trades: the limitations on the evolution of phenotypic plasticity and niche breadth. *Am. Nat.*, 148, S65–S77.

Wickham, H. (2016). *ggplot2: Elegant Graphics for Data Analysis*. Springer-Verlag New York.

Wickham, H., François, R., Henry, L., & Müller, K. (2019). *dplyr: A Grammar of Data Manipulation*. R package version 0.8.0.1. <https://CRAN.R-project.org/package=dplyr>

Tables

Table 1. Species list of *Daphnia* populations used in the Life-History Table Experiment

(LHTE)

Species	Clone	Location	Habitat Type
D. magna	MA1	Munich, Germany	Semi-permanent lake (Roulin et al. 2013)
D. magna	MA2	Tvärminne, Finland	Ephemeral rockpool (Roulin et al. 2013)
D. magna	MA3	South Dakota, US	Shallow, permanent lake (Weider et al. 2004)
D. obtusa	OB1	Oklahoma, USA	Pond
D. obtusa	OB2	Illinois, USA	Pond
D. obtusa	OB3	Missouri, USA	Pond
D. pulex	PX1	Illinois, USA	Shallow pond (Lynch 1987)
D. pulex	PX2	Illinois, USA	Shallow pond (Lynch 1987)
D. pulex	PX3	Illinois, USA	Shallow pond (Lynch 1987)
D. mendotae	ME1	Minnesota, USA	Permanent lake
D. mendotae	ME2	Minnesota, USA	Permanent lake

Table 2. Factorial MANOVA scores. Main effects and two-way interactions from a Factorial MANOVA are shown here. “Food quality” indicates the main effect of the food treatment manipulation (high phosphorus - HiP/low phosphorus - LoP). “Species” indicates the main effect of species on the response variable. “Clone” indicates the level of effect at the clonal-level, nested within species, on the response variable. Body length (mm) at the start of the experiment, the mother of the experimental animals, and time blocks were used as covariates. Two-way interactions were also tested.

Source of variance	Wilks' Lambda	df1	df2	Multivariate F	partial η^2
Start length (covariate)	0.909	6	146	2.42	
Maternal effects (covariate)	0.986	6	146	0.343	
Time (covariate)	0.925	6	146	1.983	
Food quality	0.150	6	146	137.409***	0.85
Species	0.064	18	413	37.863***	0.60
Clone	0.210	42	688	6.463***	0.23
Species * Food	0.167	18	413	20.250***	
Clone * Food	0.195	42	688	6.843***	

*** p < 0.0001

Figure Legends

Figure 1. Boxplots of life-history traits across species and across food quality treatments. Asterisks indicate significant differences between treatments were detected using linear regressions, with adjusted significance levels ($\alpha = 0.002$). Life-history traits in this study include length at maturation (A), age at maturation (B), length at the end of the experiment (C), mean length of clutches (D), mean clutch size (E), and the number of clutches (F).

Figure 2. The relationship between intraspecific trait variation (COVs) and species-level responses to food quality (Pearson correlation, $r^2 = 0.89$, $t = 9.60$, $df = 22$, $p < 0.01$).

Appendix D

RSOS-191024 Minor revisions

Associate Editor Comments to Author (Dr Michael Tobler):

The manuscript was re-assessed by two reviewers and both generally agree that the author has adequately revised the manuscript. They provide some additional feedback that should help to improve the manuscript. I would particularly like to echo two reviewer comments, one pertaining the clarification of why a negative correlation between mean and variance are expected, and one pertaining the removal of gray background in the figures. In the context of the latter, I should add that color figures are free, and using color may help to make figure 2 more clear.

I have clarified this section of the introduction as to why I would expect a negative correlation between trait variance and the expected amount of change in the trait. I do rely heavily on life-history theory which does look at trait variation and sensitivity specifically. I am extrapolating that species-level responses would be reflected by sensitivity in this case (which my equation for species-level responses also reflects). But I have also added the alternative hypothesis as suggested (see line by line comments below for more details).

I have also changed figure 2 as the two authors suggested. I now have a white background and used color to improve the figure's readability.

Please see line by line comments for each reviewer below.

Reviewer comments to Author:

Reviewer: 1

Comments to the Author(s)

The authors present a revised version of their manuscript in which they explore interspecific variation in life history traits in response to nutritional environments. They have done well to eliminate the PC analyses from the manuscript and focus instead on MANOVA analyses, which strengthen the overall statistical approach. The findings are generally expected, but interesting. The overall comparative approach is a strength.

Most of my critiques at this point are minor (see below). I would, however, like to see some effort spent in clarifying the expectation that mean and variance in responses should be inverse correlated. Indeed, my expectation for many biological traits is the opposite: that mean and variance in responses are generally positively correlated. This ends up bearing somewhat heavily on the manuscript since this prediction is introduced in the introduction and its implications are discussed in the discussion.

Thank you for reviewing this manuscript again, I really appreciate the time you put into this. Please see my general comments to the editor for how I addressed your specific comment on the correlation between species-level mean responses and intraspecific trait variation.

Line-by-line edits

Pg 5, Line 3: delete 'large'

Edited as suggested.

Pg 5, Lines 31-34: Alternatively, mean and variance is often positively correlated in biological data. Part

of this is a ‘floor effect’, for example. I would suggest discussing the alternative hypothesis (that mean and variance are positively correlated) here.

Thank you for pointing out that I should have an alternative hypothesis, especially as I find evidence for an alternative. I was not able to find a reference that directly has both species-level responses and intraspecific trait variation in Daphnia, but I provide a reference that would reflect the positive correlation between intraspecific trait variation and shifts in mean trait values.

Pg 6, Line 41: add ‘pond’ (?) after ‘ephemeral’

Added as suggested.

Pg 7, Line 26: what volume of algae (at 1 mg C L^{-1}) was fed to the zooplankton?

The volume varied (between ~500-800 microliters for HiP and ~500-1,200 microliters for LoP) depending on the concentration of algae that day, The methods that explain how to calculate this volume will give repeatability to the study. The volume was calculated each day using a spectrophotometric method that converted absorbance of chlorophyll to concentration of carbon. I modified the methods to make this more clear.

Pg 9, Lines 3-4: Missing word(s) here?

Added “than” for clarification. “...less than five percent...”

Pg 9, Line 26: change ‘was run’ to ‘was conducted’ – also change this in future sentences, only because ‘was run’ sounds a bit awkward

Edited as suggested.

Pg 10, Line 54: Please make explicit the direction of effect here – larger species have greater degrees of variation?

Added as suggested. “with larger species showing more variation than smaller species”

Pg 11, Lines 36-38: I suggest changing this heading so that it reads ‘[...] and species identity explained greater trait variation than genotype’, or something along those lines.

Edited as suggested.

Pg 11, Line 43: Change ‘imposes’ to ‘is imposed’

Edited as suggested.

Pg 14, Line 12: Missing date for reference

Thank you for pointing this out. I was missing the correct citation in this reference. This has been updated.

Figure 1: When abbreviating for microns, use mu instead of the letter ‘u’, in y-axis labels

Edited as suggested.

Figure 2: I suggest having a white background for this figure given that some of the points are quite light gray.

Edited as suggested.

Reviewer: 3

Comments to the Author(s)

I have a few minor comments that should be addressed for more transparency.

Introduction:

P5 L16 “therefore clonal variation may significantly add to genetic contributions”. The meaning of this sentence is cryptic, can you try to reformulate?

Edited as suggested.

Methods

P6 L40 I think one word is missing in the sentence “finnish pond from a ephemeral with dessication”

Edited as suggested (pond was added after ephemeral).

“incidental ambient lighting”: I suppose not only the stock culture but also the experiment were in these conditions.

Added that experimental animals also received ambient lighting for clarification.

When was the experiment conducted?

The starting date for the experiment were added for clarity.

Please provide more details about spectrophotometer measurements: wavelength, and perhaps a reference to the literature on this already established procedure?

Added reference as suggested and also added spectrophotometer details and which wavelengths were measured.

Please provide details about the experimental animals. Especially in smaller species, a single clutch contains less than 20 neonates, so to reach the indicated number clutches must have been pooled. Were all experimental animals born on the same day from different mothers, or were they all from the same mother but born on different days? I am aware this is a logistical issue but also important when interpreting results, because it influences the observed variance.

Animals were pooled from multiple mothers, and accounted for in the MANOVA. I have added a line in the methods to specify that animals were pooled from different mothers.

P9 word missing “mortality accounts for less than five percent...”

Edited as suggested.

P13 (middle) replace “we hypothesize” with “I hypothesize”

Edited as suggested.

P14 The superflea Tessier reference is incomplete, I suppose it was a formatting error.

Thank you for pointing this out. I was missing the correct citation in this reference. This has been updated.

There is no R code available in supp Mat, please provide it.

Thank you for pointing this out. Code has been added to the zip file in the datadryad submission associated with this manuscript (with updates to figure modification).

Figure 2 is hard to read. Please consider using a white background, as grey on grey is not easy. Is RSOS having a limitation on color figures? If not I would consider using colors in this figure, because even if “only” 4 shades of grey aren’t much, it is a bit hard. I also wish the symbols were larger, and I am wondering why the symbols for mean clutch size look different in shape?

Edited as suggested. Figure 2 now has a white background, different shapes for the species and different colors for the traits. Making the symbols larger caused overlap that made interpretation near the bottom left corner difficult.